# Integrative genetic analysis illuminates ALS heritability and identifies risk genes

Salim Megat [1] ✉, Natalia Mora [2], Jason Sanogo[1], Olga Roman[1], Alberto Catanese [3,4], Najwa Ouali Alami[5], Axel Freischmidt[4,6], Xhuljana Mingaj[7], Hortense De Calbiac[7], François Muratet[8], Sylvie Dirrig-Grosch[1], Stéphane Dieterle[1], Nick Van Bakel[2], Kathrin Müller[4,9], Kirsten Sieverding[4], Jochen Weishaupt[10], Peter Munch Andersen [11], Markus Weber [12], Christoph Neuwirth [12], Markus Margelisch[13], Andreas Sommacal[13], Kristel R. Van Eijk[14], Jan H. Veldink [14], Project Mine Als Sequencing Consortium*, Géraldine Lautrette[15], Philippe Couratier[15], Agnès Camuzat[8], Isabelle Le Ber[8], Maurizio Grassano[16], Adriano Chio[16], Tobias Boeckers [3,4], Albert C. Ludolph[4,6], Francesco Roselli[4,6], Deniz Yilmazer-Hanke[5], Stéphanie Millecamps [8,17], Edor Kabashi[7,17], Erik Storkebaum [2,17], Chantal Sellier[1,17] & Luc Dupuis [1,17] ✉

Amyotrophic lateral sclerosis (ALS) has substantial heritability, in part shared with fronto-temporal dementia (FTD). We show that ALS heritability is enriched in splicing variants and in binding sites of 6 RNA-binding proteins including TDP-43 and FUS. A transcriptome wide association study (TWAS) identified 6 loci associated with ALS, including in *NUP50* encoding for the nucleopore basket protein NUP50. Independently, rare variants in *NUP50* were associated with ALS risk ($P = 3.71.10^{-03}$; odds ratio = 3.29; 95%CI, 1.37 to 7.87) in a cohort of 9,390 ALS/FTD patients and 4,594 controls. Cells from one patient carrying a *NUP50* frameshift mutation displayed a decreased level of NUP50. Loss of NUP50 leads to death of cultured neurons, and motor defects in *Drosophila* and zebrafish. Thus, our study identifies alterations in splicing in neurons as critical in ALS and provides genetic evidence linking nuclear pore defects to ALS.

Amyotrophic lateral sclerosis (ALS) is the most frequent adult-onset motor neuron disease, and leads to death within a few years after onset through progressive paralysis caused by the simultaneous degeneration of upper motor neurons, located in the motor cortex, and of lower motor neurons in the brainstem and spinal cord. ALS is clinically, genetically and pathologically associated with fronto-temporal dementia (FTD). A variety of evidence has shown that ALS is a disease with high heritability[1–3]. Twin studies[4], a prospective population-based studies[5] or register studies[6] indicated a heritability ranging from 40 to 60%, while 5–10% of ALS patients show a family history[2, 3]. Genome-wide association studies of ALS led to the identification of a hexanucleotide repeat expansion (HRE) in *C9ORF72* as a major genetic cause of ALS/FTD, and 15 genes are robustly associated with ALS in a cross-ancestry manner[7, 8], some of them having been found causative in familial ALS/FTD cases (eg *C9ORF72*, *SOD1*, *KIF5A* or *TBK1*). Despite these recent breakthroughs, a large part of heritability remains to be identified in ALS. Integration of proteome or transcriptome with genome-wide association studies (GWAS), allowing identification of genes that confer risk to disease through protein or RNA abundance respectively, have yet to be performed in ALS to illuminate the nature of ALS heritability as recently exemplified in Alzheimer's[9] or Parkinson's[10, 11] diseases.

Beyond the identification of the gene variants associated with ALS, it would be critical to understand their functional relationship to well-described pathogenic events. ALS and FTD are characterized by the widespread occurrence of cytoplasmic aggregates of TDP-43, an

A full list of affiliations appears at the end of the paper. *A list of author and their affiliation appears at the end of the paper. ✉e-mail: salim.megat@inserm.fr; ldupuis@unistra.fr

RNA-binding protein preventing aberrant inclusion of cryptic exons[12]. Interestingly, decreased TDP-43 function potentiates the inclusion of a cryptic exon in the *UNC13A* risk allele identified in GWAS, leading to decreased levels of the synaptic vesicle protein UNC13A, selectively in carriers of this allele[13, 14]. Whether other variants associated with ALS are also related to the function of TDP-43 or other RNA-binding proteins (RBP) associated with ALS such as FUS, remains to be determined. Another critical pathogenic event in ALS is the dysfunction of the nuclear pore, which contributes to familial ALS caused by *C9ORF72* HRE[15–19]. Nuclear pore dysfunction precedes TDP-43 mislocalisation in neurons of sporadic ALS patients[20], and reciprocally is exacerbated by TDP-43 and FUS aggregation[21, 22]. To date, the only genetic evidence directly linking core nuclear pore component(s) to ALS, are mutations in *GLE1*, encoding a protein involved in mRNA export and translation and associated with the nuclear pore found in ALS patients[23].

Here, we show that ALS heritability is enriched in splicing regulating variants and in binding sites of a subset of RNA binding proteins including TDP-43 and FUS. Our discovery and replication transcriptome-wide association study identify 6 loci associated with ALS, among them 3 were novel. We performed rare-variant burden analysis on this set of gene in a WGS dataset including 9,390 ALS and FTD patients and 4594 controls. Among these 6 genes, rare variants in *NUP50* were associated with ALS. Furthermore, we identified ALS patients carrying haploinsufficient mutations in *NUP50*, and demonstrated the critical role of NUP50 in neuronal survival and motor neuron function in multiple models.

## Results

### Cell type specific and molecular trait heritability of ALS

We first aimed at better characterizing the involvement of specific cell types in ALS heritability. To this aim, we applied stratified linkage disequilibrium[24] (LD) score regression (S-LDSC) to the most recent ALS GWAS data[8], and compared ALS heritability estimates to two other neurodegenerative diseases (Alzheimer's disease, AD, and Parkinson's disease, PD) and two neuropsychiatric diseases (Autism spectrum disorder, ASD and schizophrenia, SCZ). To specify cell-type specific gene expression, we used datasets of single-nuclei Assay for Transposase Accessible Chromatin sequencing (snATAC-seq) of human-derived brain samples[25], identified open-chromatin region (OCR) in 6 different brain cell types, and partitioned heritability into these cell types according to the OCR annotation (Fig. 1a). Z-scores and p-values were calculated based on the coefficient $\tau_c$ defined as the contribution of annotation $c$ to per-single nucleotide polymorphism (SNP) heritability for each annotation conditional on the baseline LD and the considered trait in order to test the unique contribution of each annotation. This analysis showed that ALS risk loci were significantly enriched for OCR in excitatory and inhibitory neurons (Bonferroni corrected *p*-value <0.05), and to a lesser extent in oligodendrocytes (FDR < 0.05), but not in microglia or astrocytes (Fig. 1a, Source Data). The relevance of our analysis was corroborated by our observation of a significant enrichment of AD-risk loci in microglia-specific OCRs (Fig. 1a) and of SCZ-risk loci in neuronal specific OCRs (Fig. 1a) consistent with AD and SCZ underlying pathophysiology.

We then sought to understand whether ALS heritability was associated with specific layers of gene expression regulation in the brain. We used quantitative trait loci (QTL) for brain expression (eQTL), DNA methylation QTL (meQTL), splicing QTL (sQTL), and previously described Histone QTL (hQTL)[26] and constructed sets of annotations for each of these molecular traits using the recently described MaxCPP approach[27]. Specifically, we computed for each cis-SNP a causal posterior probability (CPP) using Causal Variants Identification in Associated Regions (CAVIAR) fine-mapping method in the 95% credible set, predicting the influence of the SNP to each of the molecular traits studied[27]. Then, for each SNP in the genome, we assigned an annotation value (MaxCPP) based on the maximum value

of CPP across all phenotypes. We included a broad set of 75 functional annotations from the baseline LD model in most analyses. To determine whether there is a potential role of splicing in constrained genes in ALS heritability, we used the ExAC gene set, which consists of 3230 genes that are strongly depleted of protein-truncating variants[28], and constructed a new splicing QTLs annotation named (MaxCPP-ExAC), defined as the maximum CPP restricted to genes in the ExAC gene set. We partitioned disease heritability on the 5 considered molecular traits and assessed statistical significance at false discovery rate (FDR) < 0.05 after Benjamini–Hochberg correction. Importantly, this strategy yielded expected results such as the observed significant enrichment for enhancer (H3K27Ac) in Alzheimer's disease (Fig. 1b, Source Data) which is supported by a prominent role of epigenetic modification[29]. In ALS, we observed enrichment in splicing QTLs, which was also observed in PD, as previously observed[10] (Fig. 1b).

Conditional analysis on the baseline LD model and all QTLs, including sQTL-ExAC showed significant enrichment for splicing QTLs in genes depleted of protein truncated variants in PD and SCZ (Fig. 1b), and a non-significant trend in ALS (*p*-value = 0.082, Z score = 1.39, Source Data), possibly due to low sample size in ALS as compared to PD and SCZ.

To better understand the role of splicing regulation in ALS, we then investigated whether heritability could also be related to binding sites of individual RNA-binding proteins (RBP), which are critical mediators of splicing[30]. To address this hypothesis, we used annotated binding sites of 125 RBPs, including 100nt flanking sequences, as defined by the eCLIP analysis[31] to partition disease heritability using applied stratified LD score regression. The LD score regression framework allows estimation of SNP effects ($\tau^*$) standardized for comparison across different disease or trait GWAS studies while conditioning on a baseline functional annotation. Importantly, we observed significantly elevated effect size ($\tau^*$) estimates in 6 RBPs after correcting for multiple hypothesis testing (Fig. 1c, FDR < 0.05, Benjamini-Hochberg correction, Source Data). Strikingly, among these 6 RBPs, we observed target sites of TDP-43 ($\tau^* = 0.72$; p-jacknife = $2.9 \times 10^{-3}$) and FUS ($\tau^* = 0.36$, p-jacknife = $2.6 \times 10^{-3}$), that contributed significantly to ALS heritability, but also associations with novel RBPs such as KHSRP ($\tau^* = 0.52$, p-jacknife = $4.2 \times 10^{-3}$) or AQR ($\tau^* = 1.44$, p-jacknife = $9.9 \times 10^{-4}$). To determine the selectivity of this involvement of RBP dysregulation in ALS, we compared global RBP binding sites dysregulation in 11 different disease traits. To this aim, we first estimated the effect size of individual RBP for each disorder while jointly conditioning on the MaxCPP QTL-based annotations and then averaged $\tau^*$ across the 125 RBPs tested in our models. We found that ALS displays the highest average RBP effect size $\tau^*$ across the 11 traits (averaged $\tau^* = 0.506$, sd = 0.1729, Fig. 1d, Fig. S1, Source Data) suggesting a prominent contribution of RBP binding sites to ALS heritability. Overall, these data suggest that ALS heritability is significantly enriched in binding sites of a subset of RBPs, which include TDP-43 and FUS.

### Transcriptome wide association study of ALS

We then performed Transcriptome-wide association studies (TWAS) to relate genetically predicted mRNA levels and disease risk[32]. The TWAS approach uses information from the RNA expression measured in a reference panel and the ALS GWAS summary statistics to evaluate the association between the genetic component of expression and ALS status. As a reference tissue, we used the collection of brain-related tissue from the GTEx consortium ($n$ = 13 tissues)[33]. We used FUSION to estimate the heritability, build predictive models, and perform TWAS[32]. To increase the robustness of our approach, we performed a two-stage TWAS, with two distinct discovery and replication cohorts.

In our discovery TWAS, we integrated genetic data from 12,577 cases and 23,475 controls[7] to our 13 reference panels which yielded 3 genome wide significant association among them 2 known ALS loci, *C9ORF72* ($Z$ = 8.81, $p$ = $1.91 \times 10^{-18}$) and *SCFD1* ($Z$ = 5.15, $p$ = $2.5 \times 10^{-7}$)

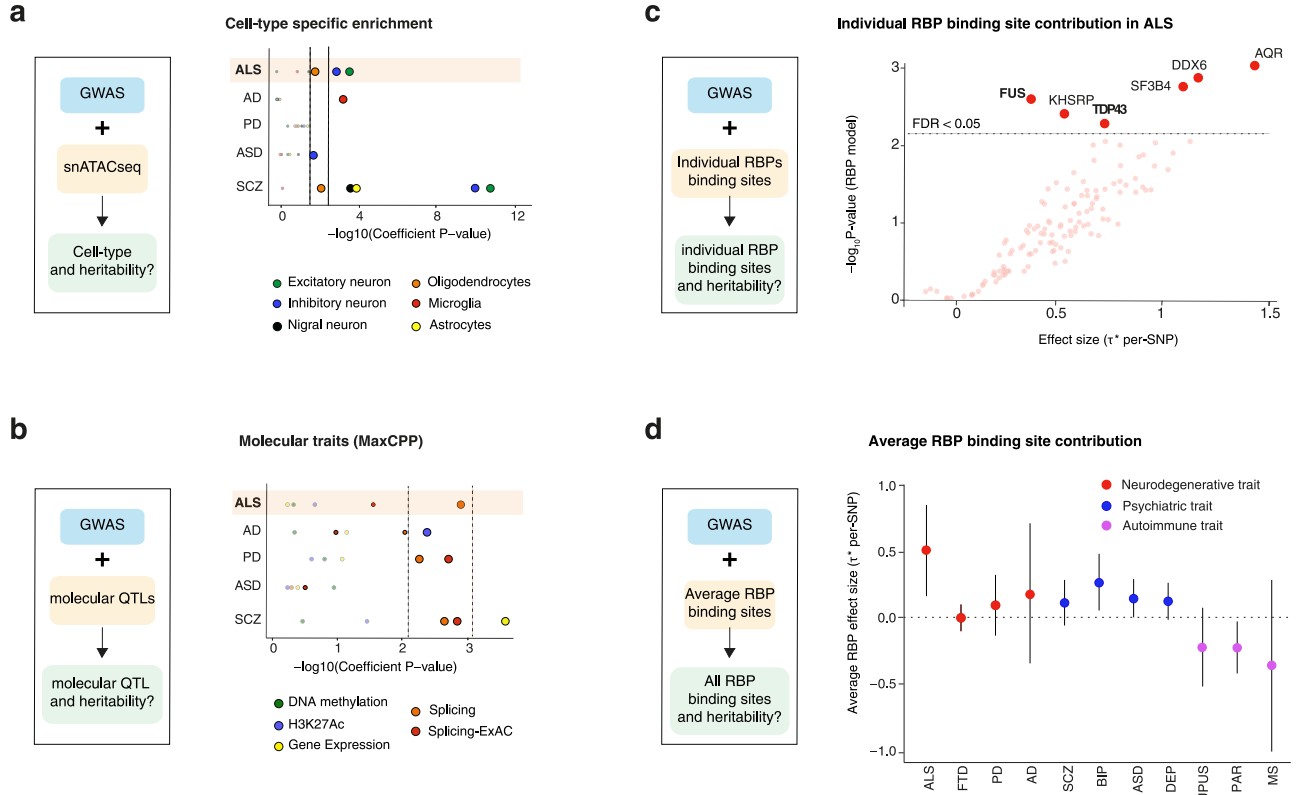

**Fig. 1 | Cell type specific and molecular trait heritability of ALS. a** Strategy to characterize cell-type specific heritability through integration of GWAS with single nuclei ATACseq (snATACseq) data (left) and *p*-values for enrichment in cell-type specific heritability in considered diseases (right). Dots indicate two-sided −log$_{10}$ (*P*-values) of enrichment obtained by linear regression model in LDSC analysis. Top dashed line indicates statistically significant enrichments after correction for multiple testing over all diseases (*n* = 5), cell types (*n* = 6) with Bonferroni correction *p* < 0.05. Bottom dashed lines indicated statistically significant enrichments after Benjamini–Hochberg multiple correction FDR < 0.05. Results indicate enrichment for inhibitory and excitatory neurons in ALS, but not in other neurodegenerative diseases. **b** Strategy to characterize molecular QTL-related heritability through integration of GWAS with QTLs data (left) and *P*-values for enrichment in heritability for 5 molecular traits in considered diseases (right). Dots indicate indicates two-sided −log$_{10}$ (*P*-values) of enrichment obtained by linear regression model in LDSC analysis. Top dashed line indicates statistically significant enrichments after correction for multiple testing over all diseases (*n* = 5), molecular QTLs (*n* = 5) with

Bonferroni correction *p* < 0.05. Bottom dashed lines indicated statistically significant enrichments after Benjamini–Hochberg multiple correction FDR < 0.05. Results indicate a global enrichment in sQTLs in ALS, PD and SCZ. Higher dashed lines indicate Bonferroni corrected *p*-values < 0.05 while lower dashed lines indicates FDR < 0.05. **c** Strategy to characterize individual RBP binding site heritability through integration of GWAS with RBP binding site data The per-SNP heritability effect sizes (τ*) for each RBP target site dysregulation is plotted for ALS GWAS. The dashed line indicates RBP models FDR < 0.05 threshold after multiple hypothesis correction (block jackknife-based one-sided *p*-values; Benjamini–Hochberg correction). **d** The per-SNP heritability effect sizes (τ*) for RBPs after conditioning on a collection of molecular QTL annotations (i.e. independent RBP effects from molecular QTLs and baseline annotations). Circle dots represent the mean of RBP effect size for each disease and error bars are 95% CI. ALS amyotrophic lateral sclerosis, FTD fronto-temporal dementia, PD Parkinson's disease, AD Alzheimer's disease, SCZ Schizophrenia, DEP depression, BIP bipolar disorder, ASD autism spectrum disorder, PAR rheumatoid arthritis, MS multiple sclerosis.

and a novel candidate gene *NUP50* (Z = −4.6, *p* = 1.94 × 10⁻⁶) (Fig. 2, Table 1, Source Data).

As a replication cohort, we aggregated genetic data for 10,035 non-overlapping cases and 16,139 controls from two recent ALS cohorts[34,35]. In this replication cohort, GWAS revealed three genome-wide significant associations in *C9ORF72* (rs700828, *p*-value = 7.2 × 10⁻⁹), *GPX3/TNIP1* (rs8177447, *p*-value = 2.74 × 10⁻⁹) and *UNC13A* (rs12608932, *p*-value = 2.94 × 10⁻⁸), which are known ALS loci. We observed moderate inflation of the test statistics ($\lambda_{GC}$ = 1.03), and linkage disequilibrium (LD) score regression yielded an intercept of 1.013 (s.e. = 0.0068), indicating that the majority of inflation was due to the polygenic signal in ALS (LD score regression (LDSC): *h*2lhl2 = 0.051, s.e. = 0.012).

We then used our replication GWAS to perform TWAS. All three TWAS associations identified in the discovery cohort were replicated in this replication cohort (*C9ORF72*, Z-meta = 10, *p* = 2.74 × 10⁻²³), *SCFD1* (Z-meta = 6.20, *p* = 5.43 × 10⁻¹⁰) and *NUP50* (Z-meta = −5.27, *p* = 1.32 × 10⁻⁷) (Source Data). Moreover, meta-analysis using a Stouffer's weighted z-score method of both discovery and replication TWAS yielded 3 novel associations such as *SLC9A8* (Z-meta = 5.31,

*p* = 1.06 × 10⁻⁷), which is a known ALS locus, *JAKMIP3* (Z-meta = 4.82, *p* = 1.36 × 10⁻⁶) and *NDUFC2* (Z-meta = −5.03, *p* = 4.88 × 10⁻⁷) (Fig. 2). To prioritize putatively causal genes, FOCUS[36] was used to assign a posterior inclusion probability (PIP) for genes at each TWAS region and for relevant tissue types. For the genomic locus 9:26112447-9:28222934, *C9ORF72* was included in the 90%-credible gene set with a posterior probability of 1 in the cerebellum (Table 1, Source Data). For the genomic locus 14:29976818-14:32381974, *SCFD1*, was part of the credible set. The highest posterior probability for causality after *C9ORF72* was 0.827 for *NUP50* in the nucleus accumbens, suggesting that *NUP50* is the causal gene in the defined region.

### Candidates risk genes associated with ALS through rare variant burden analysis

As an orthogonal approach to TWAS, we performed a rare-variant burden analysis on an additional WGS cohort of ALS-FTD patients (*N* = 2,794) and controls (*N* = 2010). In this cohort, we observed significant association with genes previously associated with FTD, such as *GRN* (*P* = 2.6 × 10⁻⁴; odds ratio = 3.6; 95%CI, 1.65–7.95) or ALS such as

**Transcriptome wide association study**

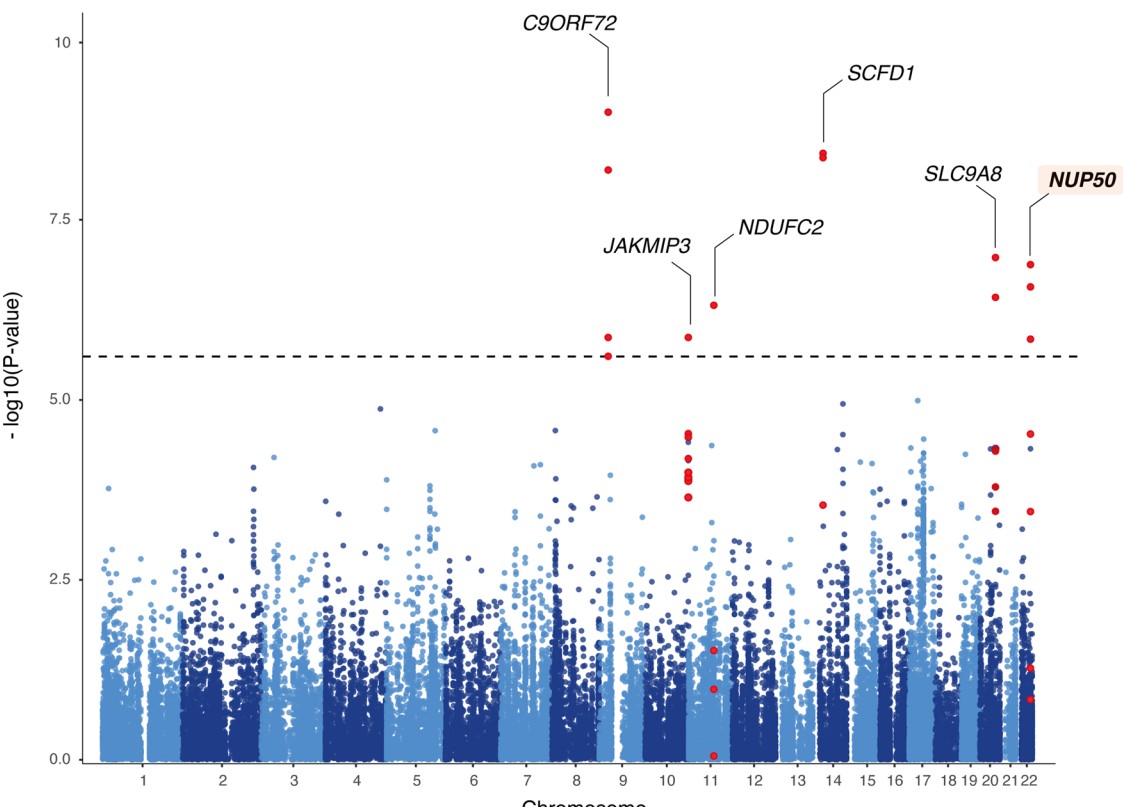

**Fig. 2 | Transcriptome-wide association study of amyotrophic lateral sclerosis.** Manhattan plot of ALS transcriptome-wide association study (TWAS) using gene expression from the GTEx consortium brain tissues. The *y*-axis corresponds to two-tailed $-\log_{10}$ (*P*-values); the *x*-axis corresponds to genomic coordinates (GRCh37). The horizontal dashed line reflects the threshold for calling genome-wide significant genes ($P = 5 \times 10^{-6}$) after Bonferroni correction.

*SOD1* ($P = 3.9 \times 10^{-4}$; odds ratio = 21.1; 95%CI, 1.15–383.3) (Fig. 3a), supporting its relevance. Among the 6 TWAS genome-wide significant genes, only *NUP50-derived* transcripts, including the canonical *NUP50* transcript (ENST00000396096), displayed a significant association ($P = 1.9 \times 10^{-3}$; odds ratio = 16.8; 95%CI, 0.90–312.2, Table 2, Source Data, $p < 0.05$ after Bonferroni Correction for 6 genes tested) for damaging variants at MAF < 0.01 (Fig. 3a).

We then sought to replicate this association of rare variants in *NUP50* in ALS using an independent whole-genome sequencing dataset 6596 ALS patients and 2584 controls (http://databrowser.projectmine.com/). Here again, in this dataset, most *NUP50* derived transcripts, including the canonical *NUP50* transcript (ENST00000396096), displayed significant association at a suggestive *p*-value ($P = 0.024$; odds ratio = 2.83; 95%CI, 1.20–6.64). Last, we performed inverse variance weighted meta-analysis using both discovery and replication cohorts and observed an increase burden of rare variant on *NUP50* ($P = 3.7 \times 10^{-03}$; odds ratio = 3.29; 95%CI, 1.37–7.87), in a range similar to *SOD1* ($P = 2.45 \times 10^{-03}$; odds ratio = 8.10; 95%CI, 1.88–34.8) in ALS patients (Fig. 3b). Interestingly, neither *GRN* ($P = 0.42$; odds ratio = 1.92; 95%CI, 0.31–11.6) nor *TET2* ($P = 0.31$; odds ratio = 1.37; 95%CI, 0.71–2.62) showed suggestive rare variant burden in the replication dataset, consistent with their association to FTD and not ALS. In all, our results demonstrate an increased burden of rare variants in the *NUP50* gene within the ALS/FTD continuum.

To characterize the potential *NUP50* variants, we filtered all SNP and short coding insertions/deletions affecting coding sequence, splice acceptor/donor site or non-coding regions (introns, 5′UTR, 3′ UTR) of *NUP50*, and identified 11 potentially pathogenic *NUP50* variants in 41 patients from both discovery and replication cohorts used in

rare variant burden analysis (Fig. 3c). Among the 9 missense and structural mutations identified across the different cohorts, all variants have CADD (combined annotation dependent depletion) scores > 20 (Table 2) suggesting high pathogenicity. For instance, we observed two rare structural variants (22:g.45564117A>G and 22:g.45567544C>T) in 12 patients but none in controls. These two variants are localized in the importin-alpha domain and predicted to impact NUP50 function (Table 2). Finally, we also observed a splice-acceptor mutation in a patient with FTD (22:g.45567480G>C). This variant has a low allele frequency (gnomAD MAF = $3.9 \times 10^{-06}$) and is predicted to disrupt a splice-acceptor site in exon 3 (Table 2). Some of the missense *NUP50* mutations identified in ALS patients were also reported at very low frequency in the ExAC database (minor allele frequency<$3 \times 10^{-5}$) classifying them as strong ALS risk factors.

We screened an additional cohort of 330 French ALS patients[37] and identified a second frameshift mutation in a patient (Patient 1) for which lymphoblasts were available. Patient 1 carried a single nucleotide deletion (variant location: 22:g.45571791delT, cDNA change:c.174delT, not found in GnomAD) leading to frameshift and premature stop codon (p.Phe58Leufs*37), which was confirmed by Sanger sequencing (Fig. S2a, b). Other ALS genes were further interrogated for this specific patient, and did not show potentially pathogenic variants, as variants in *SS18L1* (c.1036 + 17_1036 + 133del) and *NEFH* (c.85G>T, p.Ala29Ser) were present in control databases. Patient 1 had disease onset at 33 years with signs of upper and lower motor neuron involvement first restricted to the lower limbs, leading to a slowly progressive paralysis. He died from pulmonary embolism after bacterial meningitis at the age of 52 years, 224 months after the onset of the disease. Western blotting showed decreased levels of NUP50

**Table 1 | List of TWAS-identified genes**

| Gene name | CHR | START | ALS TWAS discovery 12,577 cases/23,475 controls | | | | | ALS TWAS replication 10,035 cases/16,139 controls | | | | | | |
|---|---|---|---|---|---|---|---|---|---|---|---|---|---|---|
| | | | Z discovery | P discovery | FDR | Bonferroni.P | Significant Discovery | Z replication | P.replication | Known ALS loci | Z.combined | P.combined | PIP | Causal evidence |
| C9ORF72 | 9 | 27500000 | 8.8166 | 1.180e-18 | 3.743e-14 | 4.866e-14 | YES | 4.7791 | 1.760e-06 | YES | 10.01 | 1.366e-23 | 1 | YES |
| SCFD1 | 14 | 31100000 | 5.151355 | 2.590e-07 | 6.871e-04 | 8.933e-03 | YES | 3.46811 | 5.240e-04 | YES | 6.20 | 5.438e-10 | 0.106 | NO |
| SLC9A8 | 20 | 48429250 | 3.5221 | 4.280e-04 | 5.682e-02 | 7.387e-01 | NO | 4.4035 | 1.070e-05 | YES | 5.31 | 1.061e-07 | 0.098 | NO |
| NUP50 | 22 | 45559722 | −4.76 | 1.940e-06 | 3.077e-03 | 4.000e-02 | YES | −2.3339 | 1.960e-02 | NO | −5.27 | 1.329e-07 | 0.827 | YES |
| NDUFC2 | 11 | 77800000 | −4.5597 | 5.120e-06 | 2.019e-02 | 2.624e-01 | NO | −2.194 | 2.820e-02 | NO | −5.03 | 4.885e-07 | – | NO |
| JAKMIP3 | 10 | 134000000 | 4.05207 | 5.080e-05 | 3.513e-02 | 4.567e-01 | NO | 2.6297 | 8.550e-03 | NO | 4.82 | 1.368e-06 | – | NO |

The table indicates 6 genes showing genome-wide significant association in discovery and replication ALS-GWAS. Two-sided $p$-values were adjusted at a Bonferroni $p < 0.05$ in the discovery cohort. Weighted meta-analysis of Z-scores was performed using Stouffer's method and transformed to combined two-sided $p$-values.

protein in lymphoblasts of Patient 1 as compared to 4 healthy controls (Fig. 3d, e), while *NUP50* mRNA levels were unchanged (Fig. 3f), consistent with the detection of the mutant allele expression after cDNA sequencing (Fig. S2a, b).

### Genetics of *NUP50* expression in ALS-FTD

Both TWAS and rare-variant burden analysis pointed to the possible role of the protein NUP50 in ALS pathogenesis. Supporting this, the ALS risk allele (rs6006950-G) at the *NUP50* locus was associated with decreased expression of *NUP50* in the DLPFC (rs6006950, $p = 7.5 \times 10^{-13}$), but not in the blood ($p = 0.061$) (Fig. 4a, b). Our fine-mapping method with FOCUS yielded a posterior probability of 0.82, suggesting that NUP50 is the causal gene in the defined region. These results suggest that reduced CNS expression of *NUP50* associated with this common variant could constitute a risk factor for ALS.

To further solidify the relevance of *NUP50* to ALS heritability, we studied whether the *NUP50* gene displayed binding sites of the 6 major RBPs identified in Fig. 1d. Interestingly, *NUP50* displayed binding sites for 4 out 6 of these RBPs, namely FUS, TDP-43, KHSRP and AQR (Fig. 4d). Knockdown of FUS and TDP-43, but not KHSRP or AQR, decreased *NUP50* mRNA levels in K562 cells (Fig. 4e, Source Data). *NUP50* expression was also significantly decreased in SH5SY5Y neurons and iPSC-derived motor neurons upon FUS knockdown but not TDP-43 (Fig. 4f, Source Data). Further supporting a role for *NUP50* in ALS, we observed downregulation of *NUP50* mRNA in the cortex of ALS patients (Fig. 4g, Source Data), and a trend towards decreased mRNA levels in FUS-FTD, but not TAU or TDP-43 FTD (Fig. 4g, Source Data). In addition, we observed reduced expression of NUP50 in hiPSC-derived motoneurons of patients with ALS mutations, including *FUS* ($n = 2$), *TARDBP* ($n = 2$), and *C9ORF72* ($n = 2$) (Fig. S3a, b) as well as in motor neurons of mice expressing either ALS linked mutant SOD1 or mutant FUS (Fig. S3c, d), consistent with previously published results[20, 38].

### Consequences of NUP50 depletion in cultured neurons

Reduced CNS expression of *NUP50*, either through the common risk variant or rare frameshift mutations could constitute a risk factor for ALS. We thus asked whether reducing NUP50 expression could have detrimental effects on neurons. Knockdown of *Nup50* through siRNA decreased mRNA (Fig. S4) and protein (Fig. 5a, b) levels of NUP50 in HT22 mouse hippocampal neurons, and, consistent with a function of NUP50 in nucleocytoplasmic transport, abrogated nuclear export of a fluorescent NLS-mCherry-NES reporter[39] (Fig. S5a, b). However, *Nup50* knockdown did not modify levels of other nucleoporins or of Ran and RanGAP1, two proteins critical for nuclear pore function (Fig. 5a, b). Knockdown of *Nup50* did not modify levels of endogenous TDP-43 (Fig. 5a, b) nor led to endogenous TDP-43 mislocalization (Fig. S6a, b). In addition, we did not observe modified splicing of two well documented TDP-43 splicing targets (Fig. S6c, d). However, knockdown of *Nup50* in HT22 neurons led to cytoplasmic inclusions of nuclear pore proteins, as detected using mAb414, an antibody staining multiple nucleoporins (Fig. 5c, d) and of Ran-GAP1 (Fig. S7a, b). Relevant to ALS, *Nup50* knockdown triggered also p62 positive inclusions, but not ubiquitin inclusions or stress granules (Fig. 5e, f and Fig. S7c–f). Last, *Nup50* knockdown increased neuronal death in HT22 neurons (Fig. 5g) and in primary cortical neurons (Fig. 5h). Thus, reduced NUP50 expression compromises nuclear pore function and neuronal survival in cultured neurons, recapitulating a subset of ALS pathological features.

### In vivo consequences of NUP50 depletion on motor phenotype

To further substantiate a pathogenic effect of reduced NUP50 levels in ALS, we knocked down *Drosophila Nup50*. We first evaluated the level of knock-down induced by a transgenic Nup50-RNAi line with no predicted off-targets, by ubiquitously (*actin5C-GAL4*) expressing Nup50-RNAi in third instar larvae and evaluating Nup50 transcript levels by

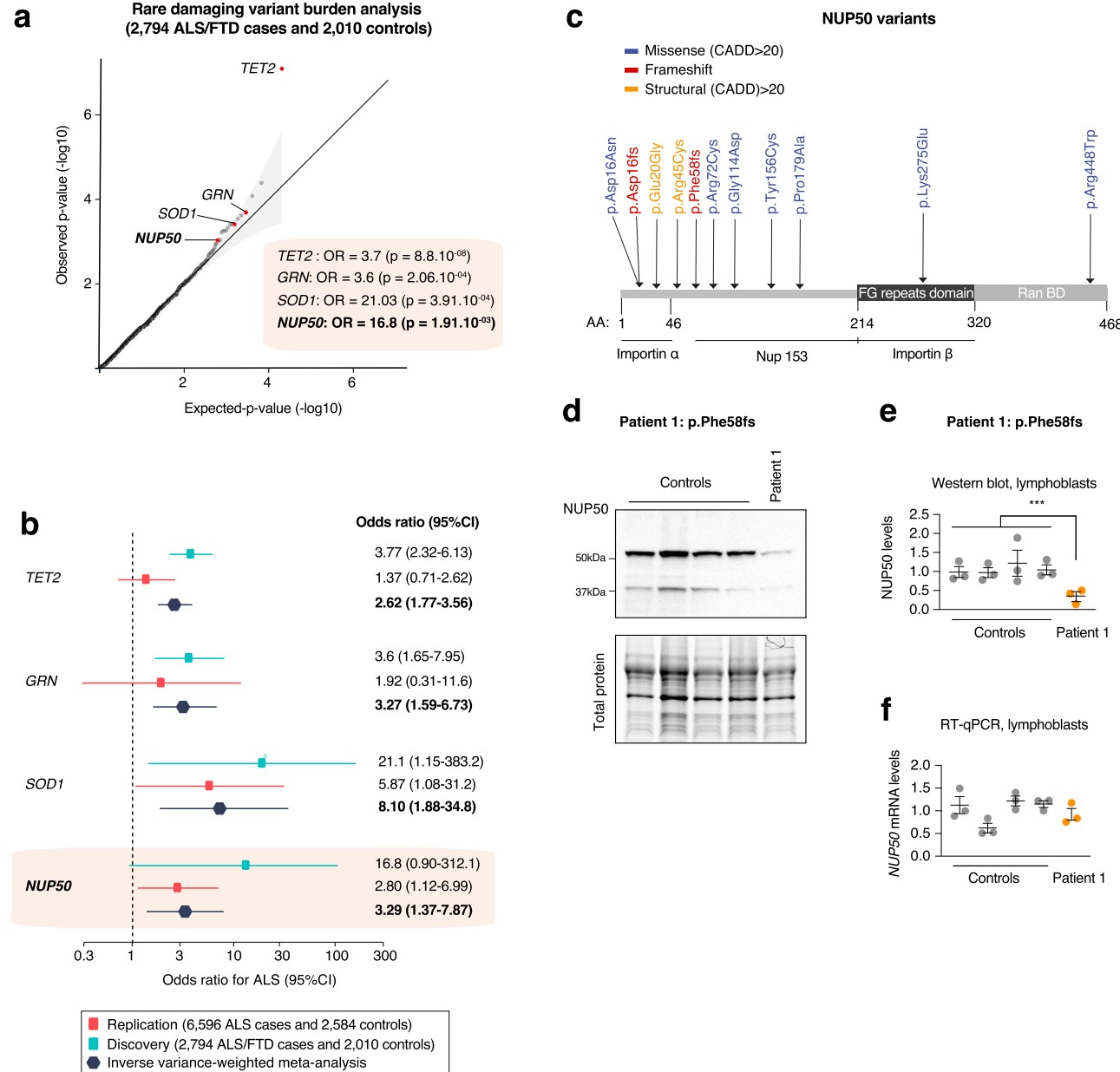

**Fig. 3 | Rare variant burden analysis prioritizes *NUP50*. a** Rare damaging variants in the replication ALS dataset. The *x*- and *y*-axis represent the negative logarithm *P* value (−log(*P*); *n* = 2794 ALS/FTD cases, *n* = 2010 controls). Q–Q plot depicting on *x*-axis the −log 10 of expected *p* value versus the actually measured *p* values from firth logistic regression for 10,182 genes. Exome-wide correction for multiple testing was set at $P < 2.5 \times 10^{-6}$, Bonferroni $p < 0.05$. Genes in red are genome-wide significant. Genes in blue are ALS/FTD known genes and candidate gene NUP50 at suggestive *p*-value $< 1 \times 10^{-3}$. **b** Analysis of candidates and ALS genes only. Forest plot display per-cohort association in discovery (6596 cases and 2584 controls) and replication (2794 ALS/FTD cases and 2010 controls). OR values and 95% CI for each cohort and meta-analysis are depicted in different color. The graphs display the means and 95% confidence interval. **c** High confidence NUP50

variants with a CADD (combined annotation-dependent depletion) score >20 are indicated. Most of these variants are located in or near the importin alpha domain of the NUP50 protein suggesting a role in nucleo-cytoplasmic transport. **d**–**f** NUP50 expression analysis in lymphoblasts from *n* = 4 healthy donors and *n* = 1 patient carrying the NUP50 frameshift Phe58fs mutation. Data are presented as mean values ± SEM *n* = 3 biologically independent experiments, each performed in triplicate. The mean value of each biological replicate is indicated by a dot on the scatter plot in (**e**, **f**). Western blotting (**b**, **c**) shows a significant decrease in NUP50 protein levels (One way ANOVA: $F_{(1,13)}$ = 12.5, $p$ = 0.00365, Control - Patient: post-hoc Tukey, adjusted-$p$ = **0.00367). However, we observed no significant changes in *NUP50* mRNA expression (**d**, Nested *t* test $t$ = 0.20955, $p$ = 0.8348).

qPCR. Nup50-RNAi expression reduced Nup50 transcript levels by 77% (Fig. 6a). Next, we evaluated the effect of *Nup50* knock-down on motor performance. We utilized an automated negative geotaxis climbing assay[40] to evaluate motor performance of 1- and 7-day-old adult flies. Whereas selective *Nup50* knock-down in motor neurons (*OK371-GAL4*) did not significantly affect motor capacities at 1 day of age, 7-day-old flies displayed a significant motor deficit (Fig. 6b, c), indicative of an age-dependent progressive motor phenotype. Finally, we evaluated

neuromuscular junction (NMJ) morphology in third instar larvae by measuring the length of the NMJs on distal muscle No. 8, as shortening of NMJ length is frequently observed in *Drosophila* models of (motor) neurodegenerative diseases[40, 41]. Motor neuron-specific *Nup50* knock-down significantly reduced NMJ length on muscle 8 (Fig. 6d–f), indicating a length-dependent NMJ phenotype.

To determine the consequences of NUP50 loss of function in a vertebrate model, we performed *nup50* knockdown by antisense-

**Table 2 | Rare variants in *NUP50* identified in ALS and controls**

| variant_type | variant_location | cDNA_change | protein_change | gnomAD_AF | ALS cases (n = 9390) | Control (n = 4594) | MVP | CADD | dbSNPid |
|---|---|---|---|---|---|---|---|---|---|
| missense_variant | 22:g.45564104G>A | c.46G>A | p.Asp16Asn | 0.000322851 | 11 | 2 | 0.87 | 26.5 | rs200329756 |
| frameshift_variant | 22:g.45564091dupATAGGAATTG | c.35_45dupATAGGAATTGG | p.Asp16fs | NA | 1 | 0 | NA | NA | NA |
| structural | 22:g.45564117A>G | c.59A>G | p.Glu20Gly | 4.2e-06 | 3 | 0 | 0.84 | 26.3 | rs1200142847 |
| structural | 22:g.45567544C>T | c.133C>T | p.Arg45Cys | 9.683e-05 | 9 | 0 | 0.89 | 28.1 | rs113634721 |
| missense_variant | 22:g.45571835C>T | c.214C>T | p.Arg72Cys | 3.23018e-05 | 1 | 0 | 0.81 | 23.6 | rs781273344 |
| splice_variant | 22:g.45574119G>A | c.341G>A | p.Gly114Asp | 0.000419897 | 3 | 0 | 0.73 | 24.2 | rs148003438 |
| missense_variant | 22:g.45574245A>G | c.467A>G | p.Tyr156Cys | 2.88367e-05 | 1 | 0 | 0.75 | 23 | rs779406443 |
| missense_variant | 22:g.45574601A>G | c.823A>G | p.Lys275Glu | 3.22768e-05 | 2 | 0 | 0.71 | 23.5 | rs753113949 |
| splice_acceptor | 22:g.45567480G>C | c.70-1G>C | NA | 3.9e-06 | 1 | 0 | NA | 32 | rs770658454 |
| missense_variant | 22:g.45574313C>G | c.535C>G | p.Pro179Ala | 6.52768e-06 | 1 | 0 | 0.81 | 22.4 | rs763689432 |
| missense_variant | 22:g.45580471C>T | c.1342C>T | p.Arg448Trp | 6.52768e-05 | 1 | 0 | 0.91 | 24 | rs777952476 |

The table indicates the type of variant, location, cDNA and protein changes, frequency in GnomAD, number in ALS and control cases and respective Missense Variant Pathogenicity prediction (MVP) score, Combined Annotation Dependent Depletion (CADD) scores and dbSNP Id of the identified *NUP50* variants in the discovery and replication cohorts.

mediated oligonucleotides with the morpholino moiety in zebrafish embryos. *NUP50* has a single zebrafish orthologue (*nup50*; NM_201580.2) with 54% target identity to the human gene. We targeted the initial AUG of *nup50* by designing a specific antisense oligonucleotide morpholino (AMO). We did not observe any significant developmental deficits or non-specific toxicity, as shown at 48 hours post fertilization (hpf) (Fig. 6f, g). However, only upon knockdown of *nup50*, we observed reduced swimming bouts in 50 hpf embryos (Fig. 6h); and analysis of the touch-evoked escape response (TEER) showed that *nup50* knockdown by AMO led to impaired locomotion with embryos displaying significantly reduced velocity (Fig. 6i). Importantly, we observed a significant reduction of zebrafish embryos displaying motor features upon co-expression of *nup50* knockdown alongside NUP50 human cDNA (hDNA) as compared to *nup50* AMO (Fig. 6g). Similarly, analysis of the velocity from the TEER swimming bouts of *nup50* AMO + NUP50 hDNA showed no difference with the control conditions (Fig. 6i).

We then analyzed motor neuron axonal projections using the *Hb9:GFP* transgenic line. We observed that motor neurons display abnormal axonal branching with reduced length upon *nup50* knockdown as compared to control conditions (Fig. 6j, k). Significantly, co-expression of NUP50 human DNA alongside nup50 knockdown also rescued the deficits observed in axonal projections from spinal motor neurons, thus confirming the specificity of the phenotype of nup50 AMO embryos. (Fig. 6j, k). These results further indicate that NUP50 loss of function is associated with motor deficits and specific alterations of the motor neuron axonal projections.

## Discussion

Our current study provides evidence that ALS heritability lies in neuronally expressed genes, and in splicing events related to a small set of RNA binding proteins. Integrating RNA expression levels in GWAS led us to prioritize 6 genes as associated with ALS risk with genome-wide significance, 3 of them in known ALS loci. Among these 6 genes, *NUP50* was the only gene also showing significant association in two independent cohorts of ALS/FTD or ALS patients using rare variant burden analysis. 3 independent biological validations strengthened that loss of NUP50 could be functionally involved in ALS. These major results are summarized in Fig. 7.

We first characterized ALS heritability by integrating GWAS with various datasets. Here splicing QTLs variants were significantly associated with ALS risk, in particular in genes expressed in neurons. However, we show here additionally that ALS risk is enriched in binding sites of TDP-43 and FUS, and 4 other RNA-binding proteins. How could this be related to disease pathogenesis? TDP-43 modulates splicing, in particular inclusion of cryptic exons. For instance, loss of TDP-43 leads to the inclusion of cryptic exons in *STMN2*[42,43] or *UNC13A*, two constrained genes critical for neuronal function. Recently, the risk allele of *UNC13A*, a major ALS GWAS signal, was found to synergize with loss of TDP-43 function to include a cryptic exon and decrease UNC13A levels[13,14]. Thus, ALS heritability enrichment in sQTLs could lead to the loss of function of these genes upon aggregation or loss of function of TDP-43, FUS or other functionally related RBPs. Further research should focus on the possible relationships between identified sQTLs and RBP dysfunction.

We then prioritized ALS genes according to predictable RNA expression levels using TWAS. Our stringent two-step TWAS strategy restricted prioritized genes to 6, with 3 novel genes. Importantly, out of these 6 genes, only *NUP50* showed an increased burden in rare variants in our discovery RVBA cohort, that was replicated in the replication cohort. These converging genetic evidence lead us to prioritize *NUP50* for functional analysis. *NUP50* encodes a protein involved in the nuclear pore complex, specifically located in the nuclear pore basket. Other research also involved NUP50 in gene expression[44] or response to DNA damage[45]. Interestingly, NUP50 has

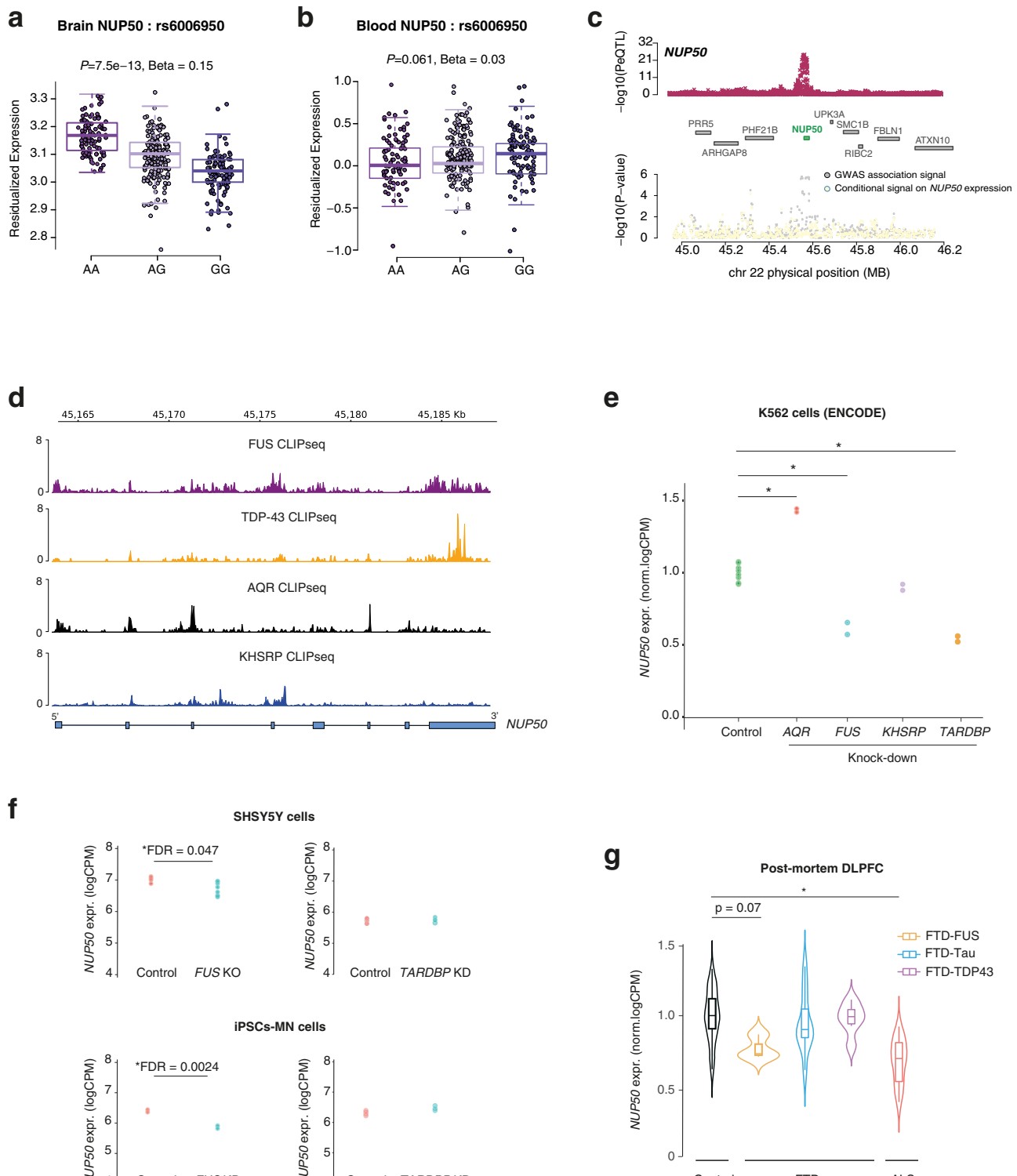

been previously associated to ALS pathogenic processes. Loss of *NupSO* enhanced the toxicity of GGGGCC repeats in *Drosophila*[15], but suppressed the toxicity of a PR25 di-peptide repeat[18] or mutant TDP-43[21, 46]. Our study provides two additional genetic evidence linking NUP50 and ALS. First, a common variant causing decreased expression of *NUP50* is associated to ALS.Second, rare variants are enriched in *NUP50* and, at least some of them lead to loss of the protein (Fig. 7).

What could be the consequences of *NUP50* loss in ALS? *Nup50* knock out mice die *in utero* due to severe neural tube defects[47]

suggesting a critical function of NUP50 in the central nervous system. Consistently, *Nup50* knockdown in primary neurons or in HT22 cell lines increased neuronal death, and knock-down of *Nup50* in Drosophila motor neurons led to a mild loss of motor function accompanied by decreased neuromuscular junction size. This is in agreement with previous studies showing normal motor neuron development in *Drosophila*[21] upon *Nup50* loss, as the defect appeared later in life, consistent with ALS adult onset. Similarly, *nup50* knockdown in zebrafish led to a reduction of evoked swimming bouts, without any apparent developmental deficits, and was associated with abnormal

**Fig. 4 | Genetics of *NUP50* expression in ALS-FTD. a, b** Boxplot showing the association between a single-nucleotide polymorphism (SNP) (rs6006950) that tags the ALS genome-wide association studies (GWAS) risk loci at *NUP50* and gene expression level of *NUP50* in DLPFC (**a**) and blood from the GTEx consortium (**b**). Each dot represents an individual patient carrying the corresponding rs6006950 SNP allele combination. rs6006950 is significantly associated with the expression level of *NUP50* in DLPFC, but not in blood. Linear regression between each genotype and residualized NUP50 expression generates *p*-values reported in (**a**) (beta = 0.15; $p = 7.5 \times 10^{-13}$) and (**b**) (beta = 0.02; $p = 0.061$). Boxplot shows median and quartile distributions, the upper and lower lines representing the 75th and 25th percentiles. **c** LocusZoom style plot for the region surrounding *NUP50* shows colocalization of the DLPFC *NUP50* expression quantitative loci (eQTL) (top panel) and ALS GWAS association signal (bottom). ALS TWAS signal at the *NUP50* locus (gray) and TWAS signal after removing the effect of *NUP50* expression (yellow). This analysis shows that the association is largely explained by *NUP50* expression.

**d** Intersection between the 6 RBPs associated with ALS and NUP50 genes coordinates identifies 4 RBP binding sites. Normalized CLIPseq tracks shows RNA-binding of 4 RBPs on the *NUP50* gene. **e** Differential expression analysis of NUP50 mRNA in K562 cells (ENCODE consortium) upon AQR, FUS, TDP-43 and KHSRP knockdown. * indicates genome-wide significant changes at FDR < 0.05 (Source Data).
**f** Differential expression analysis of *NUP50* mRNA in neuronal cell lines SHSY5Y and iPSCs-derived motorneurons upon *TARDBP* and *FUS* knockdown. *p*-values were adjusted for multiple comparison FDR < 0.05. * indicated genome-wide significant changes in genes expression at FDR < 0.05. **g** Differential expression analysis of *NUP50* mRNA of ALS-FTD patients in post-mortem DLPFC tissues. * indicates genome-wide significant change in gene expression at a FDR < 0.05. *n* = 23 controls, *n* = 3 FTD-FUS, *n* = 12 FTD-Tau, *n* = 6 FTD-TDP43 and *n* = 22 ALS. *p*-values were adjusted for multiple comparison FDR < 0.05. Boxplots show median and quartile distributions, the upper and lower lines representing the 75th and 25th percentiles.

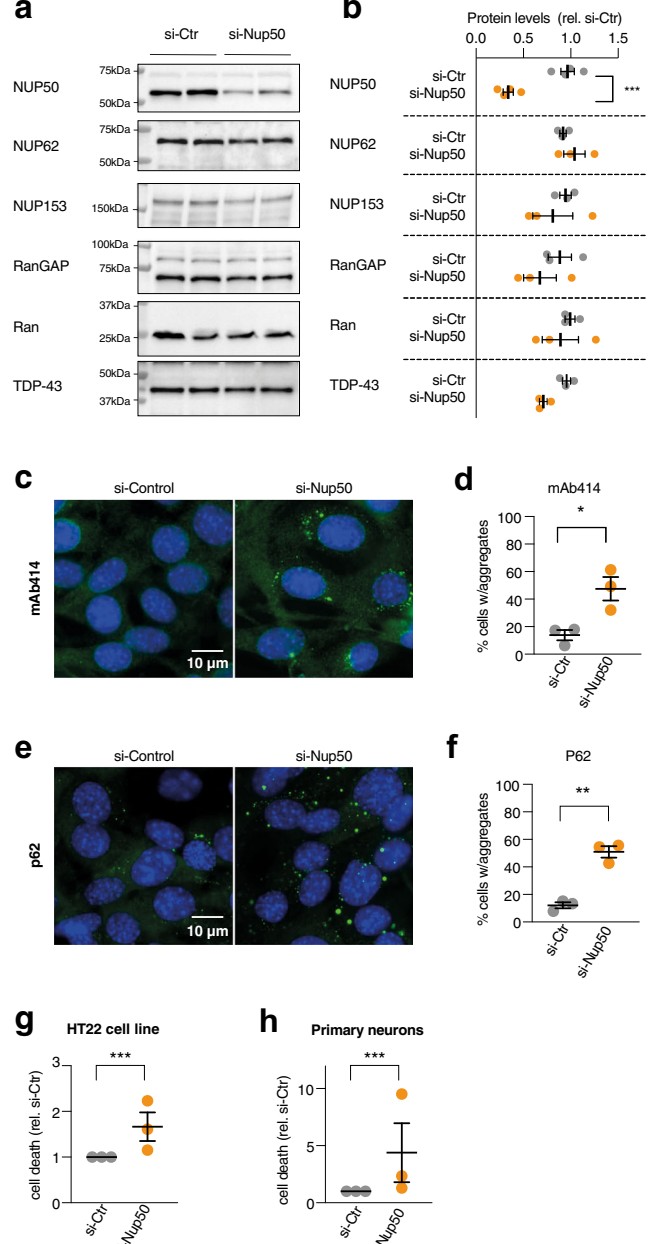

**Fig. 5 | knockdown of NUP50 in mouse neuronal cells. a** Representative images of western blots for NUP50, different nucleoporins and TDP-43. Western blots were processed in parallel to avoid cross reaction of similar-sized proteins, and quantification of the studied protein was normalized with StainFree loading control for each gel, as provided in Source Data. n = 3 biologically independent experiments, each performed in duplicate. **b** Dot plots showing a decrease in NUP50 levels (Two-tailed Nested *t* test: *t* = 8124, df = 14, ****$p < 0.0001$) but not other associated nucleoporins, RanGAP or TDP-43 ($p > 0.05$) after knock-down of the *Nup50* mRNA (si-Nup50) compared to the control condition. **c, d** Representative images and dot plots showing an increase in cytoplasmic inclusions of nucleoporins as stained with mAb414 recognizing the repeated FXFG repeat sequence in nucleoporins in HT22 cell line upon Nup50 knock-down (Two-tailed Nested *t* test: *t* = 6,778, df = 16,****$p < 0.0001$). **e, f** Representative images and dot plots showing an increase in p62 inclusion in HT22 cell line upon Nup50 knock-down (Two-tailed Nested *t* test: *t* = 9 846, df = 17,****$p < 0.0001$). **g, h** Dot-plots showing a significant increase in neuronal death (**g**) in HT22 cell lines (Two-tailed Nested *t* test: *t* = 3721, df = 24, **$p = 0.0011$) and in mouse primary neurons (**h**) (Two-tailed Nested *t* test: *t* = 3,18, df = 34, **$p = 0.0031$) For all panels, data are presented as mean values ± SEM. *n* = 3 independent experiments performed at least in duplicate. Each dot in the scatter plot indicates the mean of an individual experiment. All experiments were performed 24 h after transfection.

motor neurons. Thus, the loss of function of NUP50 is toxic to motor neurons, and sufficient to lead to motor neuron disease.

Loss of NUP50 could lead to multiple cellular consequences, that could each contribute to motor neuron demise. Indeed, *Nup50* knockdown triggered cytoplasmic inclusions of nuclear pore components, as well as of RanGAP1, a key protein regulating nucleocytoplasmic shuttling function, as observed in ALS patients[16, 21], and impaired nuclear pore function. Importantly, Coyne et al. recently showed that nuclear pore alterations were sufficient to lead to TDP-43 dysfunction in human neurons and ALS patients[20, 38]. Thus, lower levels of NUP50 might by themselves contribute to nuclear pore dysfunction, and, in turn, to TDP-43 dysfunction. In our cell models however,acute *Nup50* knockdown was not sufficient to trigger aggregation or mislocalization of endogenous TDP-43. Our results also suggest that NUP50 levels could conversely be modulated by TDP-43 or FUS dysfunction. Indeed, *NUP50* mRNA binds TDP-43 and FUS, and its levels are regulated by them. Consistently, *NUP50* is less expressed in the ALS cortex and multiple models of ALS[38]. It is thus tempting to speculate that TDP-43 or FUS dysfunction could further enhance NUP50 loss in ALS patients, in a vicious pathogenic cycle. More research is needed to characterize the pathogenic relationships between NUP50 and TDP 43 or FUS aggregation and dysfunction in ALS.

Summarizing, our current studies show that ALS is mostly associated with variants affecting splicing in neuronally expressed genes, and a number of new loci associated to ALS identified here. We further validate these genetic discoveries by validating the functional effect of reduced *NUP50* expression as a contributing factor in ALS, thus providing a genetic link between ALS and nuclear pore defects.

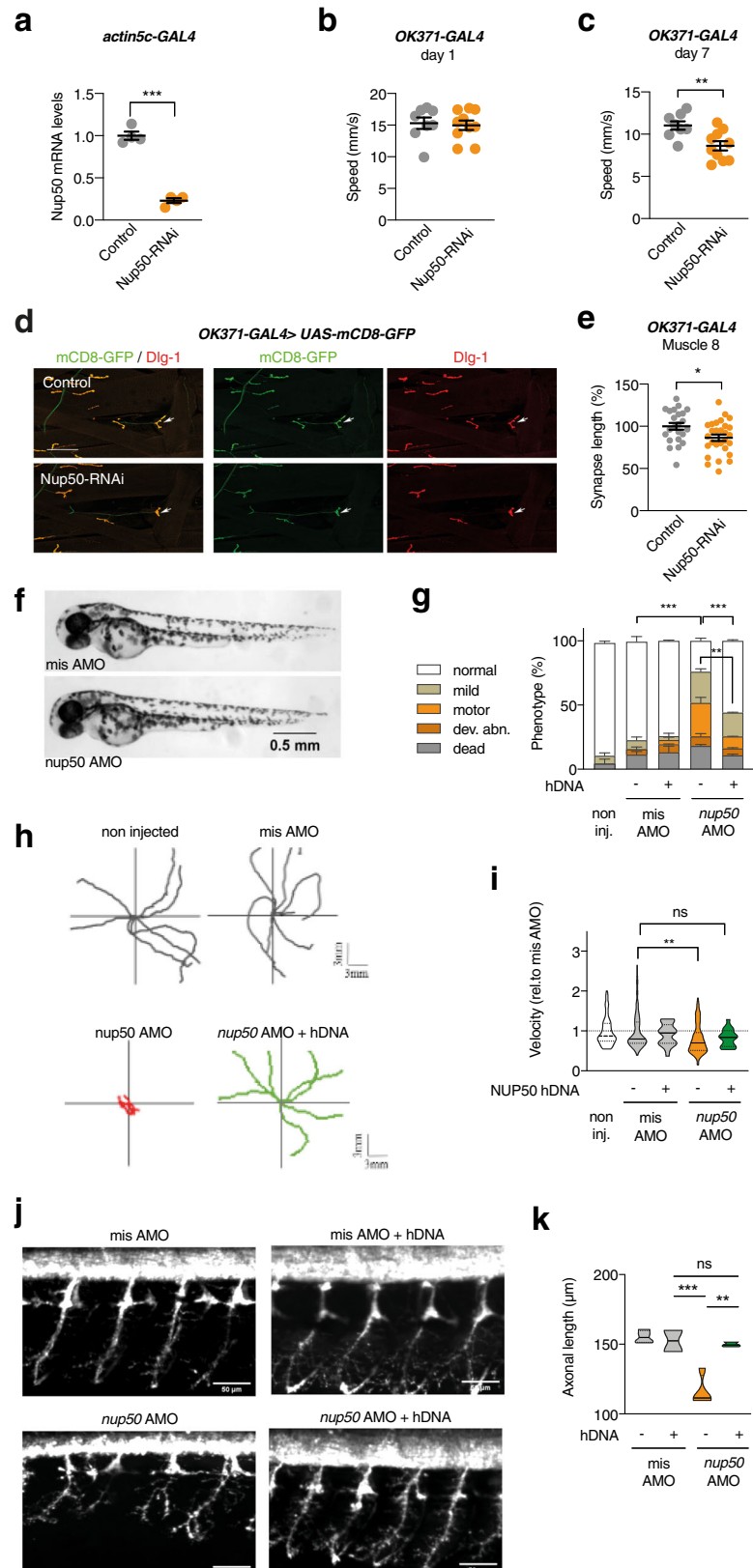

## Methods
### Ethical approval
**Patients.** French cohorts (Fig. 3, Fig. S2), protocol for genetic analyses and gene expression analysis in patient lymphoblasts were approved by the Medical Research Ethics Committee of "Assistance Publique Hôpitaux de Paris" (#A75). All FALS and ySALS patients signed a consent form for the genetic research. Autopsied patients were enrolled in the NeuroCEB brain donation program declared to the Ministry of Research and Universities, as requested by French Authorities (#AC-2013-1887). An explicit consent was signed by the patient himself, or by the next of kin, in the name of the patient, in accordance with the French Bioethical Laws. This document includes consent for

**Fig. 6 | knockdown of Nup50 leads to motor defects in Drosophila and Zebrafish. a** *Drosophila* Nup50 mRNA expression level (% driver-only control) in third instar larvae ubiquitously (*actin5C-GAL4*) expressing Nup50-RNAi. $n = 4$ per genotype; ***$p < 0.0001$ by two tailed unpaired $t$ test. Data are presented as mean values ± SEM **b**, **c** Climbing speed in automated negative geotaxis assay of 1- (**b**) or 7-day-old (**c**) male flies expressing Nup50-RNAi in motor neurons (*OK371-GAL4*). Independent groups of 10 flies: $n = 8$ (control); $n = 10$ (*OK371-GAL4*) **$p = 0.0061$ by unpaired two tailed $t$ test. Data are presented as mean values ± SEM **d** Representative images of the NMJ on muscle 8 (white arrow) visualized by membrane GFP (*UAS-mCD8-GFP*) in motor neurons (*OK371-GAL4*) labeling axons and presynaptic compartment of NMJs (green), and dlg1 (postsynaptic compartment of NMJs, red) in third instar larvae expressing Nup50-RNAi in motor neurons (*OK371-GAL4*) compared to driver-only control. Scale bar: 100 μm. **e** Synapse length on distal muscle 8 of third instar larvae expressing Nup50-RNAi selectively in motor neurons (*OK371-GAL4*). $n = 23–28$ per genotype; *$p < 0.05$ by unpaired two-tailed $t$ test. Data are presented as mean values ± SEM. **f** Representative images of 48 hpf zebrafish embryos showing no difference on

global morphology. **g** Distribution of motor phenotypes of 48 hpf zebrafish embryos among the different conditions. Data are presented as mean +sem; **$p < 0.01$ and ****$p < 0.0001$ by Tukey's multiple comparison test following 2way ANOVA. Number of independent embryos per condition: $n = 82$ non-injected, $n = 56$ misAMO, $n = 53$ misAMO+hDNA, $n = 87$ nup50 AMO, $n = 54$ nup50 AMO + hDNA in two different crossings. **h, i** Representative swimming trajectories of 48 hpf zebrafish embryos upon TEER test (**h**) and quantification (**i**). Each dot represents one embryo. Data normalized to mis AMO control are presented as violin plot. Solid line: median; dashed lines: quartiles. **$p < 0.01$ and non-significant (ns) by One-way ANOVA followed by Tukey's multiple comparison test. $n = 92$ non injected, $n = 83$ misAMO, $n = 24$ misAMO+hDNA, $n = 111$ nup50 AMO, $n = 19$ nup50 AMO + hDNA. **j, k:** Representative images of spinal motor neurons of 48 hpf zebrafish embryos (**j**) and corresponding quantification. Data are presented as violin plot. Solid line: median; dashed lines: quartiles. **$p < 0.01$, ***$p < 0.001$ and non-significant (ns) by Tukey's multiple comparison test following 1way ANOVA. $n = 4$ misAMO, $n = 2$ misAMO+hDNA, $n = 4$ nup50 AMO, $n = 3$ nup50 AMO + hDNA.

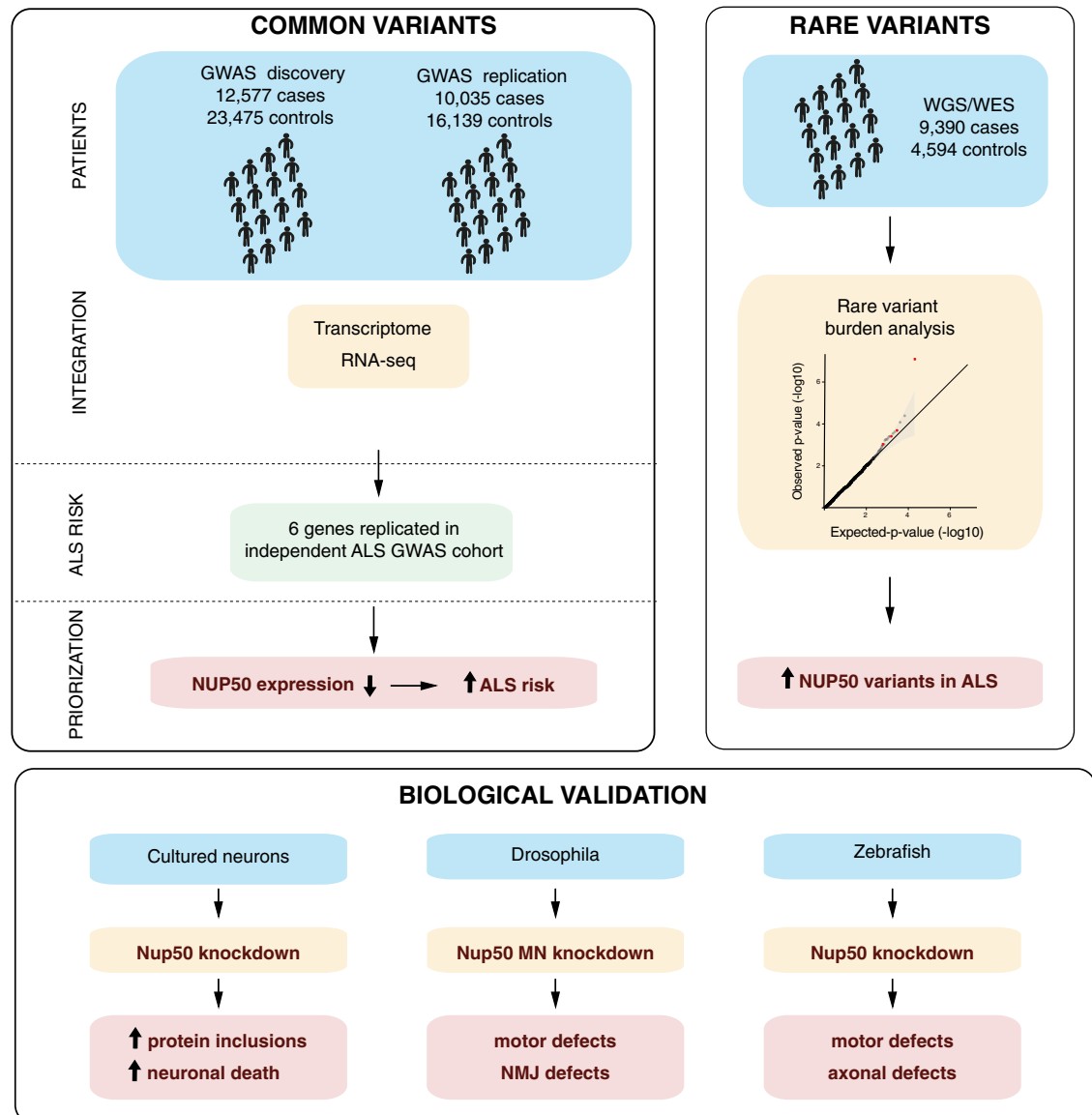

**Fig. 7 | summary of the major results of the study.** The three major results of our study are graphically summarized. See discussion for details.

genetic analyses. The diagnosis of ALS and FTD was based on published criteria.

For the German cohorts (Fig. 3, Fig. S3), all procedures with human materials were in accordance with the ethical committee of

Ulm University (Nr.19/12 and Nr. 135/20) and in compliance with the guidelines of the Federal Government of Germany. All participants or their next of kin gave informed consent for the study. The use of human material complied with the Declaration of Helsinki concerning

Ethical Principles for Medical Research Involving Human Subjects, and experiments were performed according to the principles set out in the Department of Health and Human Services Belmont Report.

**Animal models.** Transgenic mice were housed in the Faculty of medicine from Strasbourg University, 21 °C housing, 12/12 h of light/dark cycle and unrestricted access to food and water. *Fus*$^{\Delta NLS/+}$ knock-in mice have been generated in the Institut Clinique de la Souris (ICS, Illkirch, Strasbourg) and were previously described[48, 49]. This mouse strain expresses a truncated FUS protein that lacks the PY-NLS, which is encoded by exon 15 of the Fus gene and were maintained in congenic C57Bl6/J. *Sod1*$^{G86R}$ mice (FVB-Tg(Sod1*G86R)M1Jwg/J Jax Strain #:005110) were maintained in a pure FVB/N background[50]. Mouse experiments were approved by local ethical committee from Strasbourg University (CREMEAS) under reference numbers 2016111716439395 and 25452.

Adult and larval zebrafish (*Danio rerio*) were maintained at the Imagine Institutes (Paris) fish facilities and bred according to the National and European Guidelines for Animal Welfare. Experimental procedures were approved by the National and Institutional Ethical Committees (Université de Paris). Experiments were performed on wild type and transgenic embryos from AB strains as well as Hb9:GFP zebrafish allowing the observation of motor neurons axonal arborization within a somatic segment in fixed and live animals at 48–50 h post-fertilization larvae.

### Genetics

**GWAS summary association statistics.** We used the summary association statistics from the latest GWAS of ALS[8]. Also summary association statistics for SCZ and ASD were obtained from the PGC consortium (https://www.med.unc.edu/pgc/). AD and PD summary association statistics were obtained from the latest GWAS[51, 52]. All datasets used in the current study are referenced in Supplementary Data 1.

**Quality control and imputation.** For each cohort, SNPs were first annotated according to dbSNP150 and mapped to the hg38 reference genome. All multi-allelic and palindromic (A/T or C/G) SNPs were excluded. Low quality SNPs and genotyped individuals were excluded using PLINK 2.0 (–geno 0.02 and–mind 0.1). The following filter criteria were applied: MAF > 0.01, SNP genotyping rate > 0.98, Deviation from Hardy-Weinberg disequilibrium in controls $P > 1 \times 10^{-5}$. Then, more stringent QC thresholds were applied to exclude individuals: individual missingness > 0.02, inbreeding coefficient |F| >0.2, mismatches between genetic and reported gender, and missing phenotypes (PLINK–mind 0.02,–het,–check-sex). Duplicate individuals were removed (king-cutoff = 0.084). Population structure was assessed by projecting 1000 G principal components (PCs) and outliers from the European ancestries population were removed (>4 SD on PC1-4). Finally, samples in common between the individual genotype data and van Rheenen's study were identified using the checksum program id_geno_checksum and were removed from our analyses. Finally, 8214 cases and 14,129 controls were used for imputation.

**Post Imputation quality control.** Cohorts were then imputed using the TOPMED reference panel (hg38) on the TOPMED imputation server (https://imputation.biodatacatalyst.nhlbi.nih.gov/#!). Data was phased using Eagle 2.3. Post-imputation variant-level quality control included removing all monomorphic SNPs and multi-allelic SNPs from each cohort. SNPs with MAF < 1% in the TOPMED imputation panel were excluded. Subsequently, INFO scores were calculated for each cohort based on dosage information using SNPTEST v2.5.4-beta3. Within each cohort, SNPs with an INFO-score < 0.3 and those deviating from Hardy–Weinberg equilibrium at $P < 1 \times 10^{-6}$ in control subjects

were removed. All cohorts were combined including SNPs that passed quality control in every cohorts yielding 9,625,489 autosomal SNPs.

**Whole genome sequencing.** Whole genome sequencing from 3015 ALS/FTD patients and 2,031 controls were processed as described above. Duplicate individuals were removed (king-cutoff = 0.084). Population structure was assessed by projecting 1000G principal components (PCs) and outliers from the European ancestries population were removed (>4 SD on PC1-4). In total, 2794 ALS/FTD cases and 2,010 controls pass quality check analysis and were used for rare variant burden analysis.

**Association testing and meta-analysis.** After quality control, a null logistic mixed model was fitted using SAIGE with principal component (PC)1–PC10 as covariates. The model was fit on a set of high-quality (INFO > 0.9) SNPs pruned with PLINK 2.0 ('–indep-pairwise 50 25 0.2') in a leave-one-chromosome-out scheme. Subsequently, a SNP-wise logistic mixed model including the saddlepoint approximation test was performed using genotype dosages with SAIGE. To assess any residual confounding due to population stratification and artificial structure in the data, we calculated the LDSC intercept using SNP LD scores calculated in the HapMap3 CEU population.

**Fixed-effect meta-analysis of brain QTL effect sizes (FE-meta-Brain).** Meta-analysis was performed via fixed-effect model using an adaptation of the metareg function in the gap package in R. Given a set of effect sizes for SNP $i$ ($\beta_1$, $\beta_2$,... $\beta_s$) for $s$ studies, where $\beta_j$ is the eQTL effect size for study $j$, we used fixed-effect meta-analysis (FE-meta-Brain) to compute inverse-variance weighted meta-analysis $z$-scores $z_{FE}$ as follows:

$$\text{zFE} = \frac{\sum_{j=1}^{s} wj\beta j}{\sqrt{\sum_{j=1}^{s} wj}} \tag{1}$$

where $wj = \frac{1}{s.e.(\beta j)2}$ and s.e.($\beta_j$) is the standard error of $\beta_j$. We note that Eq. (1) is equivalent to computing a weighted average of $z$-scores[53]. After filtering, for low expressed genes, we used ~3 million eQTLs in 13965 genes, ~3.3 million sQTLs in 15463 genes, ~660.000 mQTLs in 46656 CpG sites.

**Brain QTLs-based annotations.** We construct the annotations for any given Brain QTL datasets using the observed marginal association statistics as previously described[53]. Briefly, each annotation is a vector that assigns a value to each SNP. Given **a** indicate our annotation for one QTL dataset where $a_j$ indicates the value assigned to SNP $j$. For continuous probabilistic annotations (MaxCPP), $0 \leq a_j \leq 1$. Let $S = (s_1, s_2, ...s_g)$ indicate an $m \times g$ matrix of the observed marginal association statistics obtained for each QTL dataset, where $m$ is the number of SNPs and $g$ is the number of eGenes (for example, genes that have at least one significant *cis*-eQTLs). Let $s_i$ be the vector of marginal association statistics of gene $i$ for all cis variants. Utilizing $s_i$ and the LD structure, we can compute the CPP for each variant. CPP is the probability that a variant is causal. Given $\alpha_{ji}$ the posterior probability that SNP $j$ is causal for gene $i$. We obtained the CPP values from CAVIAR[27]. In addition to the CPP values, CAVIAR provides a 95% credible set that contains all of the causal variants with a probability of at least 95%. We constructed the MaxCPP annotation for SNP $j$ by computing the maximum value of CPP over all genes for which SNP $j$ is in the 95% credible set of gene $i$. More formally, we have: $a_j = \max_i \alpha_{ji}$ where the maximum is over genes $i$ with $\theta_{ji} = 1$.

**Estimating the RBP dysregulation GWAS effect sizes.** We use the previously published statistical framework of stratified LD score regression (PMID 25642630) to estimate the RBP dysregulation effect

sizes for each examined disease GWAS. From the summary statistics of a GWAS, we can write the expected χ2 value for SNP j as:

$$E\left[X_j^2\right] = N\sum_c \tau_c\, l(j,c) + Nb + 1 \qquad (2)$$

where N is sample size and the annotation specific "LD score" l(j, c), representing annotation (c)'s cumulative effects tagged by the SNP j, can be written as:

$$l(j,c) = \sum_k a_c(k) r_{jk} \qquad (3)$$

where $a_c(k)$ is the annotation value at SNP k (e.g RBP dysregulation level or molecular QTLs), $r_{jk}$ is the correlation between SNP j and k in the reference panel (selected to best match the GWAS cohort), and b measuring the confounding bias[79]. Lastly, $\tau_c$ and the final standardized form $\tau_c^*$ normalized by the total SNP-based heritability and s.d. of an annotation−represents the estimated effect size of the annotation[79].

$$\tau_{rbp}^* = \frac{M^* sd_{rbp}}{h2} * \tau_{rbp} \qquad (4)$$

For RBP dysregulation annotations, τ* represents the per-SNP heritability associated with a standard deviation increase of variant RBP effect ($sd_{rbp}$). We fit $\tau_{rbp}^*$ by conditioning on a baseline annotation. The final reported RBP effect sizes ($\tau_{rbp}^*$) were jointly fit, iteratively for each RBP, with all baseline annotations.

**LD-score regression and partitioned heritability for ALS**. After obtaining the MaxCPP annotations, we ran S-LDSC[54] to generate the LD score of each variant in each annotation using the same procedure[27]. Regression SNPs, which are used by S-LDSC to estimate τ from marginal association statistics, were obtained from the HapMap Project phase[55]. Using the LD score for each annotation and the marginal statistics obtained from the trait phenotypes (GWAS-ALS), we computed the enrichment, the coefficient τ and calculated the p-value for each annotation conditional on the baseline model v1.1 and all MaxCPP annotation[56]. For cell-type enrichment analyses[25] we used human-derived single-cell ATAC sequencing data on major brain cell-types (GSE147672). Disease-relevant gene-sets annotation (MaxCPP-ExAC) were constructed as previously described[53]. Briefly, eQTLs and sQTLs annotation were restricted to the 3,230 genes that are strongly depleted of protein-truncating variants[28].

**TWAS studies**. TWAS is a powerful strategy that integrates SNP-expression correlation (*cis*-SNP effect sizes), GWAS summary statistics and LD reference panels to assess the association between the *cis*-genetic component of expression and GWAS. TWAS can leverage large-scale RNA-seq and/or methylation data to impute tissue-specific genetic expression levels from genotypes (or summary statistics) in larger samples, which can be tested to identify potentially novel associated genes[32]. Transcriptomic panels for TWAS are published[33]. We used the FUSION tool to perform TWAS for each transcriptome reference panel. The first step in FUSION is to estimate the heritability of each feature (gene expression or intron usage) using a robust version of GCTA-GREML[57], which generates heritability estimates per feature as well as the likelihood ratio tests *p*-value. We used pre-computed TWAS weights from the GTEx consortium for 13 brain tissues (http://gusevlab.org/projects/fusion/). TWAS was performed for each tissue and significant association were considered for Bonferroni *p* < 0.05 / (number of genes tested * number of tissue tested).

**Genic burden association analyses**. To aggregate rare variants in a genic burden test framework we used the method described in the ALS 2021 GWAS[8]. In short, a variety of variant filters was applied to allow for different genetic architectures of ALS associated variants per gene as was used previously[58, 59]. In summary, variants were annotated according to allele-frequency threshold (MAF < 0.01 or MAF < 0.005) and predicted variant impact ("missense", "damaging", "disruptive"). "Disruptive" variants were those variants classified as frame-shift, splice-site, exon loss, stop gained, start loss and transcription ablation. "Damaging" variants were missense variants predicted to be damaging by seven prediction algorithms (SIFT, Polyphen-2, LRT, MutationTaster2, Mutations Assessor, and PROVEAN). "Missense" variants are those missense variants that did not meet the "damaging" criteria. All combinations of allele frequency threshold and variant annotations were used to test the genic burden on a transcript level in a Firth logistic regression framework where burden was defined as the number of variants per individual. Sex and the first 20 principal components were included as covariates. All ENSEMBL protein coding transcripts for which at least five individuals had a non-zero burden were included in the analysis. Meta-analysis was performed using an inverse variance weighted methods[60].

## Whole exome sequencing and genetic analyses in French patients

Whole Exome Sequencing (WES) analyses were performed on French patient cohorts (including 200 family members with 150 index cases and 180 sporadic cases with 100 patients with ALS onset before 40 years and 80 autopsied cases) as previously described[37]. These patient cohorts were devoid of mutation in the 4 major ALS genes (including repeat expansions in *C9orf72* and/or mutation in *SOD1*, *TARDBP* and *FUS*). These exome databases were interrogated to select heterozygous or homozygous qualifying variants in coding or non coding region of *NUP50* gene with a minor allele frequency (MAF) threshold <0.005% in gnomAD database for heterozygous variants and ≤1% for homozygous variants.

Any patient with a *NUP50* variant was further interrogated for a list of 35 rare ALS related genes (including *ALS2, ANG, ANXA11, ATP13A2, CCNF, CHCHD10, CHMP2B, DAO, DCTN1, DNAJC7, ERLIN2, FIG4, GLE1, GLT8D1, HNRNPA1, HNRNPA2B1, KIF5A, MAPT, MATR3, NEK1, OPTN, PFN1, PRPH, SETX, SIGMAR1,* SPG11, *SQSTM1, SS18L1, TAF15, TBK1, TIA1, TUBA4A, UBQLN2, VAPB* and *VCP*) to select variants with a MAF <0.005% in gnomAD database. The *NUP50* frameshift variant was validated using Sanger analysis with BigDye chemistry as recommended by the supplier (Applied Biosystems).

### RNAseq analysis

The preprocessing, gene quantification and differential gene expression analysis was performed with the ARMOR workflow. All downstream analyses were performed in R and the edgeR package was used for differential gene expression analysis. We filtered the lowly expressed genes and kept all genes with a CPM of at least 10/median_library_size*1e6 in 2 replicates. Additionally, each kept gene is required to have at least 15 counts across all samples. The filtered set of genes was used for the PCA plot and differential gene expression analysis.

### Histology in mouse models

**Tissue collection and processing**. Mice were anesthetized with intraperitoneal injection of 100 mg/kg ketamine chlorhydrate (Imalgène 1000®, Merial) and 5 mg/kg xylazine (Rompun 2%®, Bayer), and then transcardially perfused with cold PFA 4% in phosphate buffered saline (PBS). After dissection, spinal cords were post-fixed for 24 h and then included in agar 4% and serial cuts of 40 μm thick were made using vibratome (Leica Biosystems, S2000). All mice studied were males.

**Immunohistochemistry**. Free-floating sections were pre-treated 30 min in citrate buffer 0.1 M pH 6.0 at 80 °C, rinsed with PBS 1X, then incubated 30mn with blocking solution (5% Horse Serum

Albumin, 1% TritonX100 in PBS). Sections were incubated overnight at 4 °C with primary antibodies diluted in PBS + 0.1% tritonX100. The following antibodies were used: rabbit anti-Nup50 antibody (Ab 137092 Abcam, 1/200) and goat anti-choline acetyl transferase (AB144P, Millipore, 1/100). After 3 rinses in PBS, sections were incubated for 1 h at room temperature with Hoechst (Sigma, B2261, 1/50.000) and secondary antibody: Donkey anti-rabbit Alexa-488 (JacksonImmunoResearch, 711-547-003, 1:1000) and donkey anti-goat Alexa-594 (Invitrogen, A11058, 1:1000). Finally sections were subsequently washed with PBS (3 × 10 min) and mounted in Aqua/polymount (Polysciences, 18606).

**Confocal microscopy and analyses.** Z-Stack images (1 μm optical section, frame resolution of 1024 × 1024 pixels) were acquired using a laser scanning microscope (confocal LSM 800 Zeiss) equipped with ×40 oil objective (NA1.4) zoom x 0.5. Excitation rays are sequential diodes 488 nm, 561 nm, 405 nm. Emission bandwidths are 400-500 nm for Hoechst, 500–570 nm for Alexa488 and 570-617 nm for Alexa594. Each Stack of 10 μm were aplane and analyze using ImageJ freeware. First, the user defined mean intensity of Nup50 in motoneurons nucleus (ChAT positive cells) for 6 spinal cords per FUS or WT genotype or 3 spinal cords per SOD or WT genotype. These values are divided in 5 categories of pixels intensity (0–50, 51–100,101–150,151–200, 200–255) and used to calculate nuclear Nup50 levels.

### Drosophila experiments

Flies were housed in a temperature-controlled incubator with 12:12 h on/off light cycle at 25 °C. OK371-GAL4 (Bl 26160), UAS-mCD8-GFP (Bl 5137) and Act5C-GAL4 (Bl 4414) were obtained from the Bloomington Stock Center (Bloomington, IN). The UAS-Nup50-RNAi line was obtained from the Vienna Drosophila Resource Center (VDRC) KK library, stock number 100564. This line is predicted to have a single ON target (Nup50) and no OFF targets.

To assay motor performance, newly eclosed male flies were collected and divided into groups of 10 individuals. Until measurement, flies were maintained at 25 °C with a 12-h light/dark cycle on standard Drosophila medium. At 1 and 7 days of age, flies were evaluated in a rapid iterative negative geotaxis assay (RING-assay), which had been previously established in the Storkebaum lab[40]. The assay is based on the innate escape response of flies to climb up the wall of a vial after being tapped down to its bottom. Flies were transferred into test tubes without anesthesia and three iterative measurements of at least 8 groups of 10 flies per genotype were video recorded with a Nikon D3100 DSLR camera. The resulting movies were converted into 8-bit grayscale TIF image sequences with 10 frames/s. Subsequently, image sequences were analyzed using an MTrack3 plug-in that automatically imports images in ImageJ, subtracts backgrounds, and filters and binarizes images to allow tracking of flies. The average climbing speed (mm/s) of all tracked flies was determined, averaged per test tube, and compared between genotypes.

To analyze synapse length on muscle 8, third instar larvae were dissected in HL3 buffer and fixed in Bouin's for 3 min. After permeabilization and blocking (10% goat serum), immunostaining was performed with anti-Discs large 1 (anti-dlg1; DSHB, 1/200). Images were taken of muscle 8 in abdominal segment 5 using a Leica SP8 laser scanning confocal microscope with 20x Plan-Apochromat objective (0.8 NA). Maximum intensity projections of z-stacks comprising the entire NMJ were used to measure the synapse length.

### Zebrafish experiments

Experiments were performed on wild type and transgenic embryos from AB strains as well as Hb9:GFP zebrafish allowing the observation of motor neurons axonal arborization within a somatic segment in fixed and live animals. Zebrafish were raised in embryo medium: 0,6 g/

L aquarium salt (Instant Ocean, Blacksburg, VA) in reverse osmosis water 0,01 mg/L methylene blue. Experimental procedures were approved by the National and Institutional Ethical Committees. Embryos were staged in terms of *hours post fertilization* (hpf) based on morphological criteria and manually dechorionated using fine forceps at 24 hpf. All the experiments were conducted on morphologically normal embryos.

Morpholino antisense oligonucleotides (AMOs; GeneTools, Philomath, USA) were used to specifically knockdown the expression of the *nup50* orthologue in zebrafish (NM_201580.2) at the final concentration of 1 mM. The AUG Morpholino oligonucleotide (AMO) sequence targeting nup50 from 5' to 3' and complementary to the translation-blocking target is the following: GGCCATCA-CAGCTCAACTGAACACC, binding the mRNA target [in brackets]: cttaccagtg tgtgtgacgc gaacggttgg cgggaaac[gg tgttcagttg agctgtg(atg) gcc]aagcgga ttgcggaaaa ag. A linearized plasmid containing human NUP50 cDNA tagged by mCherry was used for the rescue experiments at the final concentration of 100 ng/μL.

The locomotion of 50 hpf zebrafish embryos was measured using the *Touched-Evoked Escape Response* (TEER) test. Embryos were touched on the tail with a tip and the escape response were recorded using a Grasshopper 2 camera (Point Grey Research, Canada) at 30 frames per second. Velocity parameter was quantified per each embryo using the video tracking plugin of ImageJ 1.53 software (Sun Microsystems, USA).

50 hpf *Hb9:GFP* zebrafish embryos were fixed in 4% paraformaldehyde and captured at the same defined four somatic segments with a Spinning Disk system (Zeiss, Germany). Average axonal lengths in these segments were quantified using ImageJ 1.53 software (Sun Microsystems, USA).

### Cell culture

**Lymphoblast cultures.** Lymphoblastoid cell lines from the French ALS patient carrying the *NUP50* frameshift variant were established by Epstein Barr virus transformation of peripheral blood mononuclear cells. Lymphoblasts from four healthy individuals (aged of 39, 50, 54, and 73 years) were used as control. Lymphoblasts were grown in RPMI 1640 supplemented with 10% fetal bovine serum, 50 U/ml penicillin and 50 mg/ml streptomycin (Life Technologies) renewed twice a week.

**Mammalian cell cultures.** Primary cortical neurons were prepared from C57Bl/6 mouse embryos at E18 and grown on polylysine-coated 24-well plates in neurobasal medium supplemented with 1 × B27, 0.5 mM L-glutamine, and 100 IU/ml penicillin/streptomycin at 37 °C with 5% $CO_2$. Neurons were seeded on 6-well plates and co-transfected at day 5 using Lipofectamine 2000 (Invitrogen) with GFP plasmid plus control siRNA (Horizon # D-001810-01-20) or siRNA NUP50 (Horizon # L-042342-01-0020) at a final concentration of 100 nM. The medium was replaced after 4 h with a 1:1 (v:v) mixture of conditioned and fresh neurobasal medium. After 24 h, determination of neuron cell death was performed by flow cytometry.

Mouse hippocampal neuronal cell HT22 (SCC129 Sigma Aldrich/Merck) were grown in Dulbecco's Modified Eagle's Medium (DMEM) 4.5 g/l glucose supplemented with 10% fetal calf serum and 1% penicillin–streptomycin at 37 °C in 5% $CO_2$. For immunofluorescence studies, HT22 cells were seeded 24 h before transfection on glass coverslips and transfected using Lipofectamine 2000 (Fisher Scientific) for NES-mcherry-NLS reporter (Addgene plasmid # 72660, a gift from Barbara Di Ventura, Roland Eils) experiments plus siRNA control (Horizon # D-001810-01-20) or siRNA NUP50 (Horizon # L-042342-01-0020) or RNAiMAX (Fisher Scientific) with control siRNA or siRNA NUP50 at final concentration of 100 nM to analyze endogenous protein localization after NUP50 depletion. For immunoblot analyses, HT22 cells were seeded on 6-well plates and after 24 h, cells were transfected using RNAiMAX (Fisher Scientific) with control siRNA

(Horizon # D-001810-01-20) or siRNA NUP50 (Horizon # L-042342-01-0020) at final concentration of 100 nM.

**Human iPSCs.** The hiPSC lines used in this study are detailed in Supplementary Data 3. The lines generated at Ulm University have been previously published[61, 62] while the other lines have been commercially purchased. hiPSC were cultured using mTeSR1 medium (Stem Cell Technologies, 83850) on Matrigel® -coated (Corning, 354277) 6-well plates using at 37 °C (5% $CO_2$, 5% $O_2$). When colonies reached 80% confluence, they were passaged in 1:6 split ratio after detachment using Dispase (Stem Cell Technologies, 07923). We differentiated MN from hiPSCs as previously described[62]. Briefly, confluent hiPSC colonies were detached and transferred ultra-low attachment T25 flasks (Corning, CLS430639) for the formation of embryoid bodies (EBs) in hESC medium (DMEM/F12 + 20% knockout serum replacement + 1% NEAA + 1% β-mercaptoethanol + 1% antibiotic-antimycotic + SB-431542 10 µM + Dorsomorphin 1 µM + CHIR 99021 3 µM + Pumorphamine 1 µM + Ascorbic Acid 200 ng/µL + cAMP 10 µM + 1% B27 + 0.5% N2). On the fourth day of culture in suspension, medium was switched to MN Medium (DMEM/F12 + 24 nM sodium selenite + 16 nM progesterone + 0.08 mg/mL apotransferrin + 0.02 mg/mL insulin + 7.72 µg/mL putrescine + 1% NEAA, 1% antibiotic-antimycotic + 50 mg/mL heparin + 10 µg/mL of the neurotrophic factors BDNF, GDNF, and IGF-1, SB-431542 10 µM, Dorsomorphin 1 µM, CHIR 99021 3 µM, Pumorphamine 1 µM, Ascorbic Acid 200 ng/µL, Retinoic Acid 1 µM, cAMP 1 µM, 1% B27, 0.5% N2). After 5 further days, EBs were dissociated into single cells with Accutase (Sigma Aldrich, A6964) and plated onto µ-Plates (Ibidi, 82406) pre-coated with Growth Factor Reduced Matrigel (Corning, 354230).

Immunostainings were performed as previously described[62]. Cells were fixed with 4% paraformaldehyde (containing 10% sucrose), and incubated for two hours using blocking solution (PBS + 10% Goat Serum + 0.2% Triton-x100; the same solution was used for the incubation with primary antibodies for 24 hours at 4 °C). Cells were than incubated with the following primary antibodies: anti-CHAT (Abcam, a custom-made version of the ab181023 antibody raised in rat; diluted 1:500), anti-NUP50 (Abcam, ab137092; diluted 1:500). After overnight incubation, three washes with PBS were performed before incubating the cells with secondary antibodies (Alexa Fluor® from ThermoFisher Scientific; diluted 1:1000 in PBS) for two hours at room temperature. Afterwards, cells were washed again 3 times and mounted with Pro-Long™Gold Antifade mountant with DAPI (ThermoFisher Scientific, P36934) mixed to ibidi Mounting Medium (Ibidi, 50001). Confocal microscopy was performed with a laser-scanning microscope (Leica DMi8) equipped with an ACS APO 63 x oil DIC immersion objective. Images were captured using the LasX software (Leica), with a resolution of 1024 × 1024 pixels and a number of Z-stacks (step size of 0.5 µm) enough to span the complete cell soma. To analyze the intensity of nuclear NUP50 (with ImageJ), we extracted the single plan from the complete Z-stack where nucleus of the neuron of interest was showing the largest shape.

**Viability assay.** For cell viability, cells were detached by scraping and resuspended in PBS. TO-PRO-3 iodide (Fisher Scientific, T-3605) was added at 20 nM to each sample and gently mixed just prior to analysis, and 50,000 cells were FACS-analyzed using BD LSRII flow cytometer (BD Biosciences).

**Immunofluorescence of HT22 cells.** For ubiquitin, RanGAP1, P62SQSTM1, Nup153, NPC, G3BP1 immunofluorescence, coverslips were incubated for 10 min in PBS with 4% paraformaldehyde, washed with PBS, and incubated in PBS plus 0.5% Triton X-100 during 10 min. The cells were washed with PBS and the coverslips were incubated during 1 h with primary antibody (1:200) against ubiquitin (3933 S, Cell signaling), RanGAP1 (330800, Thermo Fisher Scientific), p62/SQSTM1

(abcam56416) Nup153 (Abcam 24700), NPC (abcam Ab24609), G3BP1 (proteintech 13057-2-AP). After washing with PBS, the coverslips were incubated with a goat anti-mouse (ThermoFisher scientific A-11001) or anti-rabbit (ThermoFisher scientific A-11008) secondary antibody conjugated with Alexa488 for 1 h, washed twice with PBS, before mounting in FluoroMount-G mounting medium with DAPI (FisherScientific # 15596276).

## Western blotting
All uncropped western blot images are provided in Source Data.

**Western blotting of lymphoblast extracts.** Lymphoblasts cells were homogenized in lysis buffer (50 mm Tris, pH 7.4, 150 mm NaCl, 1 mm EDTA, pH 8.0, and 1% Triton X-100) containing protease and phosphatase inhibitors (Sigma-Aldrich) and protein concentration was quantitated using the BCA protein assay kit (Pierce). Fifteen micrograms of proteins were loaded into a gradient 4–20% SDS-PAGE gel (Bio-Rad, 5678094) and transferred on a 0.45 µm nitrocellulose membrane (Bio-Rad) using a semi-dry Transblot Turbo system (Bio-Rad). Membranes were saturated with 10% nonfat milk in PBS and then probed with the anti-NUP50 (Abcam, ab137092, 1:1000) primary antibody diluted in 3% nonfat milk in PBS. Blots were washed and incubated with anti-Rabbit secondary antibody conjugated with HRP (P.A.R.I.S, BI2407, 1:5000) for 2 hours. Membranes were washed several times and analyzed with chemiluminescence using ECL Lumina Forte (Millipore, WBLUF0500) using the Chemidoc XRS Imager (Bio-Rad). Total proteins were detected with a stain-free gel capacity and normalized.

**Western blotting of HT22 neuronal extracts.** Twenty four hours after transfection, HT22 cells were collected in RIPA (TrisHCl pH8 −50mM, NaCl-150mM, EDTA pH8-1 mM, 1% Triton X-100) supplemented with protease inhibitors (Complete protease inhibitor cocktail - Merck #11697498001), incubated 30 minutes on ice and centrifuged at 4 °C at 15682 g for 10 min. Eight µg of protein was denatured by heating at 95 °C for 10 min and loaded into 4-20% Criterion TGX stain free precasted (Biorad) and transferred onto nitrocellulose membrane using Trans-blot Turbo (BioRad). The membranes were incubated in 5% PBS-milk solution for 1 h, then with the primary antibody (1:1000) against ubiquitin (3933 S, Cell signaling), RanGAP1 (330800, Thermo Fisher Scientific), p62/SQSTM1 (abcam56416) Nup153 (Abcam 247000), NPC (abcam Ab24609), G3BP1 (proteintech 13057-2-AP) in PBS overnight at 4 °C, washed 3 times with PBS and incubated with anti-mouse (Abliance BI4413) or rabbit (Abliance BI2407) horseradish peroxidase-conjugated secondary antibody. After washing three times with PBS, the membranes were developed with ECL Luminata Forte Western HRP (Millipore) and scanned with a BioRad Molecular Imager ChemiDoc XRS + . Stain free detection was used as loading control and image analyses were performed using ImageLab BioRad software.

## RT-qPCR
**RT-qPCR of lymphoblast RNA.** 1 µg of RNA was reverse transcribed with iScript™ reverse transcription (Biorad, 1708841). The quantitative polymerase chain reaction was performed using Sso Advanced Universal SYBR Green Supermix (Bio-Rad) and quantified with Bio-Rad software. Gene expression was normalized by calculating a normalization factor using *ACTIN*, *TBP* and *POL2* genes according to GeNorm[63]. Primer sequences are provided in Supplementary Data 2.

**RT-qPCR of Drosophila RNA.** Total RNA was extracted from third instar larvae using Trizol reagent (Life Technologies). Following DNase treatment, the SensiFAST cDNA synthesis kit (BioLine) was used for reverse transcription to cDNA, using 2 µg RNA as starting material. Resulting cDNA samples were used as templates for real-time PCR assays performed on a BioRad CFX system using the SensiFAST SYBR

No-ROX Kit (BioLine). Primers used for quantification of Nup50 transcript levels are Actin42A mRNA was used as housekeeping gene for normalization. Data were analyzed using the ΔΔCt calculation method. Experiments included no-reverse transcriptase and no-template controls.

## Statistics

All data are presented as mean ± standard error of the mean (SEM), except otherwise stated. Differences were considered significant when $p < 0.05$. The test used is mentioned in the figure legend. For *Drosophila*, a Robust regression and Outlier removal method (ROUT) was performed to detect statistical outliers. All data points that were considered outliers were excluded from further data analysis.

## Reporting summary

Further information on research design is available in the Nature Portfolio Reporting Summary linked to this article.

## Data availability

The GWAS discovery/replication summary statistics and QTL annotation data generated in this study have been deposited in the zenodo database link: https://zenodo.org/record/7385500#.Y4h06uzMKBQ. The following publicly available datasets were used in this project: NIH Genome-Wide Association Studies of Amyotrophic Lateral Sclerosis; accession code: phs000126.v1.p1, Genome-Wide Association Study of Amyotrophic Lateral Sclerosis in Finland; accession code: phs000336, Genetic Epidemiology of Refractive Error in the KORA (Kooperative Gesundheitsforschung in der Region Augsburg) Study; accession code: phs000125.v1, International Age-Related Macular Degeneration Genomics Consortium−Exome Chip Experiment; accession code: phs000187.v1. DEMENTIA-SEQ: WGS in Lewy Body Dementia and Frontotemporal Dementia; accession code: phs001963.v2.p1, AnswerALS; accession code: https://www.answerals.org/ The single-cell ATACseq data used in this study are available under the accession: https://www.ncbi.nlm.nih.gov/geo/query/acc.cgi?acc=GSE147672
Other data generated in this study are provided in the Supplementary Information/Source Data file. Source data are provided with this paper.

## Code availability

R code and other relevant codes are posted at: https://github.com/SalimMegat/NUP50-role-in-ALS.git.

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

## Acknowledgements

The authors would like to thank Andrea CHICI (Neuromuscular Disease Unit/ALS Clinic, Kantonsspital St. Gallen, St. Gallen, Switzerland), Pierre de la Grange (Genosplice, Paris, France), Danielle Seilhean (ICM, Paris, France), Charles Duyckaerts (ICM, Paris France), NeuroCEB (Paris, France), the Généthon cell and DNA bank (Evry, France) for patient tissues, DNA and lymphoblasts, and Anca MARIAN (Imagine) for technical assistance, iGenseq and iCONICS core facilities (Paris, France) for whole exome sequencing data. Image acquisition and image analysis were performed on the Imaging Platform of the CRBS, PIC-STRA UMS 38, Inserm, Unistra. We would like to thank the DEMENTIA-Seq and the AnswerALS consortia for providing genomic data used in this manuscript. This study was funded by Agence Nationale de la Recherche (ANR-16-CE92-0031, ANR-16-CE16-0015, ANR-19-CE17-0016, ANR-20-CE17-0008 to L.D.), Fondation pour la recherche médicale (DEQ20180339179 to L.D. and post-doctoral position to SaM), Axa Research Funds rare diseases award 2019 (L.D.), Fondation Thierry Latran (HypmotALS to L.D. and F.R. and Trials to F.R.), Motor neuron disease association MNDA Dupuis/Apr16/852-791 (L.D.), Association Française contre les Myopathies (AFM-Téléthon, #19466 to StM, #23646 to L.D. and E.S.), Radala Foundation for ALS Research (2019 to F.R., 2020 to L.D. and E.S.), Deutsche Forschungsgemeinschaft (German Research Foundation) (Project-ID 251293561, CRC 1149, individual grants 443642953, 431995586 and 46067541 to F.R.), Association pour la Recherche sur la Sclérose latérale amyotrophique et autres maladies du motoneurone (ARSla, S.3200.ARSLA.1 to StM and 2022 to L.D.), University of Strasbourg Institute for Advanced Studies (USIAS) (2019 to L.D.), Joint Programme—Neurodegenerative Disease Research (JPND; grant numbers ZonMW 733051075 (TransNeuro) and ZonMW 733051073 (LocalNMD) to E.S.), European Research Council (ERC) consolidator grant (ERC-2017-COG 770244 to E.S), European Research Council (ERC) grant agreement no. 772376—EScORIAL (JHV) and Brain-Cognition-Behaviour Doctoral School, (ED3C) at Sorbonne University (FM).

## Author contributions

S.Me. and L.D. conceptualized the study. S.Me. performed statistical genetic analysis. N.M., N.v.B., and E.S. performed *Drosophila* experiments. S.Me., J.S., O.R., S.D., and C.S. performed experiments in lymphoblasts of Patient 1, HT22 and primary neurons. A.Cat. and T.B. performed experiments in iPS-derived motor neurons. N.O.A. and D.Y.H. performed pathological analysis in patients. A.F., F.M., K.M., K.S., J.W., P.M.A., M.W., C.N., M.M., A.S., G.L., P.C., A.C.L., F.R., and S.Mi. performed exome sequencing, identified and characterized NUP50 gene variants and identified Patient 1. I.L.B. and A.Cam. performed RNAseq in FTD patients. A.Ch. and M.G. provided additional WGS for RVBA. X.M., H.d.C. and E.K. performed zebrafish experiments. S.D.G. performed histology in mouse models. K.R.v.E. and J.H.V. performed genetic analysis and gene burden analysis. The work was supervised by S.Me., S.Mi., E.K., E.S., C.S., and L.D. Manuscript was drafted by S.Me. and L.D., and reviewed and accepted by all authors. N.M., J.S., O.R., A.Cat., N.O.A., and A.F. contributed equally to this study and should be considered as joint second authors.

## Competing interests

J.H.V. reports having sponsored research agreements with Biogen. L.D. reports having sponsored research agreements with Inflectis Bioscience and Cytokinetics. All other authors declare no competing interests.

## Additional information

[1]Université de Strasbourg, Inserm, Mécanismes centraux et périphériques de la neurodégénérescence, UMR-S1118, Centre de Recherches en Biomédecine, Strasbourg, France. [2]Department of Molecular Neurobiology, Donders Institute for Brain, Cognition and Behaviour and Faculty of Science, Radboud University, Nijmegen, Netherlands. [3]Institute of Anatomy and Cell Biology, Ulm University, Ulm, Germany. [4]German Center for Neurodegenerative Diseases (DZNE) Ulm, Ulm, Germany. [5]Clinical Neuroanatomy, Department of Neurology, Ulm University, Ulm, Germany. [6]Department of Neurology, Ulm University, Ulm, Germany. [7]Laboratory of Translational Research for Neurological Disorders, Imagine Institute, Université de Paris, INSERM UMR 1163, 75015 Paris, France. [8]Sorbonne Université, Institut du Cerveau - Paris Brain Institute - ICM, Inserm, CNRS, APHP, Hôpital de la Pitié Salpêtrière, Paris, France. [9]Institute of Human Genetics, Ulm University, Ulm, Germany. [10]Division for Neurodegenerative Diseases, Neurology Department, University Medicine Mannheim, Heidelberg University, Mannheim, Germany. [11]Department of Clinical Science, Neurosciences, Umea University, Umea, Sweden. [12]Neuromuscular Disease Unit/ALS Clinic, Kantonsspital St. Gallen, St. Gallen, Switzerland. [13]Institute for Pathology, Kanstonsspital St. Gallen, St. Gallen, Switzerland. [14]Department of Neurology, University Medical Center Utrecht Brain Center, Utrecht University, Utrecht, The Netherlands. [15]Service de Neurologie, Centre de Référence SLA et autres maladies du neurone moteur, CHU Dupuytren 1, Limoges, France. [16]ALS Center "Rita Levi Montalcini" Department of Neuroscience, University of Turin, Turin, Italy. [17]These authors contributed equally: Stéphanie Millecamps, Edor Kabashi, Erik Storkebaum, Chantal Sellier, Luc Dupuis. A full list of members and their affiliations appears in Supplementary Information.
✉e-mail: salim.megat@inserm.fr; ldupuis@unistra.fr

## Project Mine Als Sequencing Consortium

Jan H. Veldink ⓘ [14]

