## [Peer Review File · Nature Communications]

Integrative genetic analysis illuminates ALS heritability and identifies risk genesReviewers' comments:

Reviewer #1 (Remarks to the Author):

Alterations to the composition and function of the nuclear pore complex are widely considered a prominent pathologic hallmark of ALS. Recently, a number of studies have begun to identify genomic variants as risk factors for ALS pathogenesis. In this study, Megat and colleagues identify variants in the Nup50 gene in 23 ALS patients. They show that these variant lead to decreased Nup50 expression and that knockdown of Nup50 can lead to motor deficits in flies and fish. While this topic and pathway is of high interest to a broad readership, there are a number of substantial concerns in the current study that seriously precludes any real scientific interpretation. The data presented throughout much of the manuscript do not provide strong support for the authors conclusions. In its current form, the manuscript purely replicates prior studies identifying decreased Nup50 pathology in multiple genetic forms of ALS. Specific comments are detailed below. This reviewer is not an expert in genomic analyses and thus, the majority of the comments and evaluation of this manuscript are focused on the biological impact of Nup50 variants and decreased expression.

Major Concerns:

1. The authors largely use an antibody targeting the C-terminal region of Nup50 for their studies. While this suggests they are detecting only "normal" full length Nup50 protein, this is not confirmed. Thus, they authors must perform additional western blot and immunostaining studies with an antibody (or several antibodies) that recognize additional epitopes throughout the Nup50 protein to thoroughly evaluate which protein species result from the genetic variants (eg no protein from variant allele? "cryptic peptides" from variant allele?). Additionally, mass spec peptide identification studies could be employed. This approach is standard when examining coding variants and is absolutely required to even begin to interpret the data.
2. On lines 329-330, the authors state "immunofluorescence analysis showed decreased levels of Nup50 in neurons and widespread ring-shaped perinuclear neuronal Nup50 labeling". Authentic Nup50 should in fact be detected as a ring around DAPI signal when in central z planes as Nups localize to the edge of the nucleus when associated with NPCs. This is not pathology. This is normal Nup50 distribution. See also comment 3 below as this fluorescence intensity can not be directly compared to intranuclear staining observed in different planes of the nucleus.
3. For image analysis of Nup50 staining (lines 930-943), the authors state they analyze a minimum of 4-5 MNs from 5-8 sections per case. Given that they are analyzing roughly 10 um thick optical sections, these number are incredibly low- and completely unacceptable. Motor neurons have large nuclei (in the XY and Z planes) and thus it is highly unlikely the authors 1. Captured the full 3D volume of these nuclei in all sections and 2. Captured identical 10 um thick sections across all nuclei. Thus, there will be a large amount of inherent variability and these numbers are simply insufficient. Even though many of the images presented are entirely oversaturated (which also would make quantification inaccurate), it is quite easy to tell (Fig 4f) that some neurons analyzed are in completely different nuclear planes. As a comparison, the prior appropriately carried out literature and experiments on nuclear and nup abnormalities examined more than 5000 nuclei!
4. The quantification for Figure 4f should be presented in the main figure with the RT-PCR data. Further, given that the authors performed RT-PCR from whole motor cortex (4e), they should also perform western blots as was done in Fig 4b-d.
5. The authors make a point that Nup50 variants were not detected in patients with known genetic mutations. However, they then go on to stain for Nup50 in postmortem tissues, mouse models, and iPSCs with known ALS mutations. This is largely not novel as decreased Nup50 expression has previously been reported in both C9orf72 and sALS. Further, there is a huge disconnect between Nup50 variants as a risk factor and decreased Nup50 expression (eg Nup50 pathology) in the context of known genetic mutations. These are two very different questions. It is also unknown whether Nup50 variants were also present in the postmortem tissues and iPSC lines used for these studies. Moreover, previous work has suggested that in both C9orf72 and sALS, decreased Nup50 expression follows POM121 reduction as part of a complex NPC injury cascade. Thus, simply staining for Nup50 in

the context of multiple genetic mutations simply replicates previous results and most certainly does not provide strong evidence that loss of Nup50 is a risk factor for ALS. In fact, the authors even mention in line 353 that "loss of nuclear Nup50 is a common downstream event of ALS mutations." Thus, the authors themselves contradict their own claim that loss of Nup50 is a risk factor by indicating it is a downstream event as has been previously reported!

6. In figures 5a and 5c, the authors need to comment on the variability of Nup50 staining in WT vs mutant conditions and/or provide better images and quantification. You can very clearly see that WT and mutant images presented were not taken from the same z plane (nuclear rim vs diffuse nuclear staining of Nup50, Chat observed "within the nucleus" vs completely cytoplasmic). As a result, the presented quantification appears to be an artifact of this major experimental issue.

7. Again, lines 345-364 "mice displayed abnormal Nup50 immunoreactivity characterized by a perinuclear ring". This is not abnormal. This is in fact where Nups should localize when analyzing a central z plane of a nucleus.

8. In figure 5e, the Chat and Nup50 staining need to be presented as individual images. As the figure stands now, the Nup50 signal is completely masked by Chat in some images leading the reviewer to believe that presented z plane and/or collapsed z stack is not identical across conditions and thus will also drastically effect quantification.

9. The authors should confirm their iPSN IF data by western blot for Nup50 levels given the issues mentioned above.

10. In figure 5f, C9, TDP and FUS should NOT be grouped together for statistical analyses. These are 3 very different genetic mutations and thus should be treated individually.

11. Why are the authors using mouse hippocampal neurons to demonstrate human ALS relevance of Nup50 KD? At minimum, mouse cortical neurons should also be employed. However, given that the authors clearly have access to control iPSNs, it is unclear why the experiments were not done in this more biologically relevant system.

12. Further, in reference to Figure 6, there are a number of concerns. 1. By western blot, the authors show no change in Nup62 and Nup153 levels following Nup50 KD. However, they show "mAb414 aggregates". Curiously, mAb414 recognizes Nup62 and Nup153. The authors do not report on nuclear mAb414 staining, nor do they show individual channel panels for Fig 6c making it incredibly difficult to reconcile and interpret this data. Based on the preliminary evidence shown, it would be expected that Nup62 and Nup153 levels may be increased. 2. The images presented in Fig6c are extremely poor quality. mAb414 staining in the siControl panel looks simply like background staining. There should be a distinct bright rim of immunostaining surrounding DAPI with minimal cytoplasmic staining. Thus, one has to question whether their si-Nup50 staining is simply an artifact. In Fig6h, there is quite the range of cell death. In fact, only 1 of the replicates shows increased cell death whereas the other 2 are nearly identical to controls. Thus, I question the validity of the conclusion that decreased Nup50 increases neuronal death.

13. The authors broadly assay Ran and RanGAP1 expression but also show defects in the localization of a NCT reporter upon Nup50 KD. Given that the authors highlight the interaction between Nup50 and importins earlier in the text/figures, they should additionally evaluate importin levels, function, and localization in the context of Nup50 KD.

14. The in vivo Nup50 KD studies in both flies and fish are incredibly difficult to interpret and the data presented do not support the authors claims. 1. In both models, the authors chronically KD Nup50. As NPC function and composition is critical for cell division, one can not disentangle developmental vs degenerative effects of reduced Nup50 expression. 2. In flies and fish, it is critical that the authors show that Nup50 protein was reduced. 3. In flies, ubiquitous reduction of Nup50, but not motor neuron specific depletion, resulted in motor deficits. This data alone suggests a large non-neuronal contribution of Nup50 KD to behavioral abnormalities. Thus, any NMJ or motor axon phenotypes may be the result of non-cell autonomous defects. The authors begin the manuscript by highlighting that genetic variants are found within genes presumably predominantly expressed in neurons.

15. It is unclear how the authors performed their statistical analyses. The authors need to explicitly state whether the average of all neurons analyzed from a single patient, mouse, iPSN culture etc was taken as $n = 1$ or whether each individual cell analyzed represents $n = 1$. The latter is inappropriate.

16. The manuscript lacks the critical experiments which would indicate Nup50 variant induced

decreased expression is a risk factor for ALS. It should be absolutely required that the authors generate iPSCs with specific Nup50 variants (in an otherwise completely wildtype iPSC) and evaluate Nup50 expression, and multiple ALS phenotypes and pathologies (eg nucleocytoplasmic transport, TDP-43 function and localization, neuronal survival, axon growth etc). Without these experiments, this manuscript simply identifies Nup50 variants in an ALS population and confirms previous reports of decreased Nup50 expression in familial and sporadic forms of ALS.

Minor Concerns:

1. In the results sections for their genetic analyses, it might be helpful if the authors could give a brief less technical description. Much of the technical jargon could be moved to the methods or supplement. This would make the manuscript more accessible to a broad scientific audience.
2. The authors comment that ALS risk loci were present in excitatory and inhibitory neurons and to a lesser extent oligodendrocytes (line 131-132). Could they please comment on the identity of these oligodendrocytes? For example, are they mature oligodendrocytes or OPCs? What was the marker used to identify them? Neurons and OPCs derive from the same lineage so the population that is similar to excitatory and inhibitory neurons may in fact be this common precursor cell.
3. In their exome-wide association analysis (line 286-287), the authors evaluate nearly 3X more ALS cases compared to controls. Given the genetic heterogeneity amongst the population, this seems like a disproportionately low number of controls especially when compared to earlier analytics when control number matched or exceeded ALS. Can the authors please comment on this?
4. Line 927 (Fig..)

Reviewer #2 (Remarks to the Author):

The study includes the application of transcriptome-wide association analysis to studying the genetic architecture of ALS. Using TWAS, the authors find 59 genes significantly associated with ALS. Among the newly associated genes (45), the authors observed the decreased expression of NUP50, encoding a nuclear pore complex protein. In follow-up experiments, they show decrease expression of NUP50 in two patients with NUP50 variants. In patient 1 (carrying the NUP50 frameshift Phe58fs mutation), however, decreased protein levels were not associated to decreased mRNA levels (Figure 4 legend). In patient 2 (carrying the near-splice NUP50 mutation c.1086-6C>T), assessment of NUP50 mRNA levels in motor cortex was not quantitative and the RT-PCR shown in Figure 4 (panel e), shows hardly any change in steady state NUP50 mRNA levels. The lack of changes in steady state NUP50 mRNA levels in cells with NUP50 variants seem to contradict the main assumption of the TWAS study.

Reviewer #3 (Remarks to the Author):

This manuscript from Megat and colleagues is composed by two parts.

In the first part, the latest ALS GWAS is analysed in combination with a) cell-specific ATAC-seq data to define different cell type contributions, b) eQTL, meQTL, sQTL and pQTL data to identify processes relevant to ALS (and not to other NDDs). Results are then complemented by PRS scores, from large DNA resources.

In the second part, to identify novel ALS genes, the group narrows down the investigations to the role of NUP50, a nuclear pore gene that has previously been described to be altered in ALS post mortem brains and iPSC neurons, but has never been genetically linked to ALS. Beyond identifying NUP50 as one of the genes emerging, the authors search for and find possible pathogenic variants linked to ALS, thus linking NUP50 genetics to the disease. Overall this is a vast amount of work.

Generally, it would be great if the authors could expand on their analysis of brain and patient cells in order to build more support for the pathogenicity.

The drosophila and zebrafish experiments are interesting, but also would benefit from investigations to

add support for the partial loss of NUP50 in being pathogenic.

Following are my queries:

- Protein and RNA data from Figure 4 should be harmonised. it would be important to use QPCR on brain samples (4E), as in the previous panel with lymphoblasts.
- Pathology in 4F is unclear. Some of the highlighted changes appear to be present in control as well. Quantification of features such as the ring-shaped perinuclear staining is needed in order to be convincing.
- The fact that NUP50 decreases also in control ALS is interesting and in line with previous other findings, but it does not help convince the reader about the pathogenicity of the mutation. Could the authors perform blots in other unaffected regions (eg cerebellum)? One would expect a similar decrease in the case with splicing changes, but not in the ALS control.
- Similarly, to add support to the pathogenicity claims, as Coyne et al have shown reduction in other proteins in their extensive nuclear pore characterisation, it would be very useful if investigations in the lymphoblasts and brains were extended to at least one that changes and one that seems to remain unchanged.
- In regards to the possible splicing mutant. In Supplementary figure 9 it would be reassuring to see detection of the SNPs in equal amounts using genomic DNA, though I appreciate the exact same combination of primers may not be used as these are for cDNA.
- In figure 4F, which is the ALS1 case from 4E?
- In figure 5. Intensity and pathology should be analysed at both early and symptomatic timepoints.
- Is the intensity in other cell types, eg in dorsal SC unchanged? This would help convince of the specificity.
- Similarly to above, inclusion of other nuclear pore proteins would be useful, to show whether this is specific to NUP50.
- Results should be also reported on a per animal basis, not just per cell.
- Figure 5. Very small error bars in 5F. Can results be shown with scatter plots, and can the different replicates be highlighted by colour? Is this result in line with Conye et al at earlier timepoints? If cultures are pure enough, could authors provide RNA analysis as well?
- The progressive phenotype in flies is interesting. Is there evidence of this being specific for motoneurons? Are there any pathological features that can be investigated beyond just the motor phenotype?

General comments

We observed that the vast majority of the comments of all three reviewers were restricted to commenting on the NUP50 part of the manuscript. This was likely due to our choice of title, that reflected only this part of the study. We also noted that most critical comments were rightfully focused on the pathological part of the study, that included only one NUP50 mutation carrier, for a mutation whose pathogenicity could not be easily ascribed. As this was not an essential part of the study required to convey our claims, we removed it from the study. We are currently seeking to obtain additional autopsies of other NUP50 patients that could be included in a follow up study.

With these two major observations, we decided to revise the manuscript in several directions, with the overall general goal to improve readability and focus on established substantiated claims.

The most important changes made include:

- 1) Changing the title and focus of the study, to highlight the importance of the first genetic part. We think that the new title, by providing a more general overview of the study decreases the confusion originating from the previous title and that suggested that the only important result was the discovery and characterization of NUP50. We also added a summary figure at the end of the paper in order to better explain the common and rare variants and synthesize the key findings.
- 2) Strengthening the genetic part. We have added replication of the TWAS in an additional reference panel, replicating 30 genes, as well as a second cohort of ALS-FTD patients in which we perform rare variant burden analysis, later meta-analyzed with Project Mine. We now provide (i) genome wide significant evidence for NUP50 common variants to be associated with ALS, (ii) significant rare variant burden association in two independent cohorts of ALS-FTD and ALS patients, (iii) evidence for haplo-insufficiency in patients derived cells, and (iv) functional validation in 3 independent models. This genetic part has been entirely rewritten, removing previous figure 2, and providing a number of novel complementary statistical analysis showing robustness of the approach (Figure S1, S2), relevance of the results (Figure S3)
- 3) Removing possible premature or unsubstantiated claims. Further to comments of the three reviewers, we removed figure 4e and f, and the reference to the pathology study. Previous Figure 5 has been redesigned according to their comments and is now only presented as supplementary information (Figure S8) with substantially decreased claims. We think it remains an interesting information, and maintained it in the current version.

Reviewer #1 (Remarks to the Author):

Alterations to the composition and function of the nuclear pore complex are widely considered a prominent pathologic hallmark of ALS. Recently, a number of studies have begun to identify genomic variants as risk factors for ALS pathogenesis. In this study, Megat and colleagues identify variants in the Nup50 gene in 23 ALS patients.

They show that these variant lead to decreased Nup50 expression and that knockdown of Nup50 can lead to motor deficits in flies and fish. While this topic and pathway is of high interest to a broad readership, there are a number of substantial concerns in the current study that seriously precludes any real scientific interpretation. The data presented throughout much of the manuscript do not provide strong support for the authors conclusions. In its current form, the manuscript purely replicates prior studies identifying decreased Nup50 pathology in multiple genetic forms of ALS. Specific comments are

detailed below. This reviewer is not an expert in genomic analyses and thus, the majority of the comments and evaluation of this manuscript are focused on the biological impact of Nup50 variants and decreased expression.

We thank the reviewer for acknowledging that the topic of our study is generally interesting for a broad readership. The current version of the manuscript is vastly different from the initial version of the manuscript, and many of the comments of this reviewer are referring to figures that were removed from the current manuscript. We generally agree that our characterization of NUP50 pathology could have been more thorough. However, we validated NUP50 as a model example of TWAS nominated gene. We refer this reviewer to the general comments above that details the major changes to the manuscript, made in response to reviewer's comments and our own thinking.

In general, we are surprised that a reviewer focuses his/her comments on a very tiny part of the study and does not even try to catch the big picture. Almost every comment by this reviewer are related to previous Figures 4e-f and 5, which are now either removed or reorganized and sent to supplementary information.

Major Concerns:

1. The authors largely use an antibody targeting the C-terminal region of Nup50 for their studies. While this suggests they are detecting only "normal" full length Nup50 protein, this is not confirmed. Thus, they authors must perform additional western blot and immunostaining studies with an antibody (or several antibodies) that recognize additional epitopes throughout the Nup50 protein to thoroughly evaluate which protein species result from the genetic variants (eg no protein from variant allele? "cryptic peptides" from variant allele?). Additionally, mass spec peptide identification studies could be employed. This approach is standard when examining coding variants and is absolutely required to even begin to interpret the data.

We are very surprised, and respectfully disagree with the reviewer when he/she states that it is standard to use multiple antibodies along with mass spectrometry measurement to characterize protein levels.

We have tried several commercially available antibodies for NUP50 and this antibody was the only one to reliably yield a band in western blots at the expected molecular weight.

In the previous manuscript, the NUP50 antibodies were used in Figure 4b to show the effect of the frameshift mutation. As the reviewer states this shows a decrease in the normal NUP50 protein. As the frameshifted allele (confirmed in Sanger sequencing) deletes the ORF from its 58th amino acid (out of 468), we think it is very reasonable to state that this variant is indeed a frameshift mutation. In this set of experiments, the use of additional antibodies or MS techniques will not bring much more to this notion. We never claimed that cryptic peptides would be included in the frameshift mutation carrier NUP50, and do not understand what the reviewer is referring to.

NUP50 antibodies were also used in previous figure 4f, now removed.

Previous figure 5 is now included in supplementary information (current Figure S8), as confirmation of previous studies. In these figures, the use of the NUP50 antibody was only made to claim for decreased immunoreactivity. As mice or cells under study do not carry specific NUP50 variants, there is no ground to use epitope mapping techniques for these experiments.

Last, we used this NUP50 antibody in cells to measure efficacy of knockdown (previous Figure 6a, current figure 5a). This western blot is complemented by RT-qPCR showing the

same NUP50 downregulation. Thus, we confirm here, using a completely orthogonal approach, that our knockdown strategy is efficient.

In all, we removed the data in which the comment of the reviewer on NUP50 antibody could have been relevant, and the remaining uses of NUP50 western blotting are actually standard and supporting our claims.

2. On lines 329-330, the authors state “immunofluorescence analysis showed decreased levels of Nup50 in neurons and widespread ring-shaped perinuclear neuronal Nup50 labeling”. Authentic Nup50 should in fact be detected as a ring around DAPI signal when in central z planes as Nups localize to the edge of the nucleus when associated with NPCs. This is not pathology. This is normal Nup50 distribution. See also comment 3 below as this fluorescence intensity can not be directly compared to intranuclear staining observed in different planes of the nucleus.

This comment refers to a figure (Figure 4f) that has been removed, thus addressing the concern of the reviewer.

3. For image analysis of Nup50 staining (lines 930-943), the authors state they analyze a minimum of 4-5 MNs from 5-8 sections per case. Given that they are analyzing roughly 10 um thick optical sections, these number are incredibly low- and completely unacceptable. Motor neurons have large nuclei (in the XY and Z planes) and thus it is highly unlikely the authors 1. Captured the full 3D volume of these nuclei in all sections and 2. Captured identical 10 um thick sections across all nuclei. Thus, there will be a large amount of inherent variability and these numbers are simply insufficient. Even though many of the images presented are entirely oversaturated (which also would make quantification inaccurate), it is quite easy to tell (Fig 4f) that some neurons analyzed are in completely different nuclear planes. As a comparison, the prior appropriately carried out literature and experiments on nuclear and nup abnormalities examined more than 5000 nuclei!

This comment refers to a figure (Figure 4f) that has been removed, thus addressing the concern of the reviewer.

4. The quantification for Figure 4f should be presented in the main figure with the RT-PCR data. Further, given that the authors performed RT-PCR from whole motor cortex (4e), they should also perform western blots as was done in Fig 4b-d.

This comment refers to a figure (Figure 4f) that has been removed, thus addressing the concern of the reviewer.

5. The authors make a point that Nup50 variants were not detected in patients with known genetic mutations. However, they then go on to stain for Nup50 in postmortem tissues, mouse models, and iPSCs with known ALS mutations. This is largely not novel as decreased Nup50 expression has previously been reported in both C9orf72 and sALS. Further, there is a huge disconnect between Nup50 variants as a risk factor and decreased Nup50 expression (eg Nup50 pathology) in the context of known genetic mutations. These are two very different questions. It is also unknown whether Nup50 variants were also present in the postmortem tissues and iPSC lines used for these studies. Moreover, previous work has suggested that in both C9orf72 and sALS, decreased Nup50 expression follows POM121 reduction as part of a complex NPC injury cascade. Thus, simply staining for Nup50 in the context of multiple genetic mutations simply replicates previous results and most certainly does not provide strong evidence that loss of Nup50 is a risk factor for ALS.

We acknowledge that this part of the work could be viewed as partially replicating previous work. Indeed, we cited and discussed the studies that the reviewer mentions (references 12 to 22, and 45 of the previous manuscript). We therefore strongly toned down our claims, and have put the previous figure 5 as supplementary information (novel Figure S8).

In fact, the authors even mention in line 353 that “loss of nuclear Nup50 is a common downstream event of ALS mutations.” Thus, the authors themselves contradict their own claim that loss of Nup50 is a risk factor by indicating it is a downstream event as has been previously reported!

We do not understand the point of the reviewer: there is no contradiction between the fact that genome wide significant NUP50 variant decreases NUP50 expression and increases the risk of developing the disease (what we show) and the fact that NUP50 loss can be triggered or exacerbated by currently known genetic factors. On the contrary, we would speculate that these two pathogenic events likely synergize in patients carrying both fALS mutations and the NUP50 risk allele. We never claimed that NUP50 loss causes all ALS nor that the only cause of NUP50 loss would be genetic variants (either common or rare).

6. In figures 5a and 5c, the authors need to comment on the variability of Nup50 staining in WT vs mutant conditions and/or provide better images and quantification. You can very clearly see that WT and mutant images presented were not taken from the same z plane (nuclear rim vs diffuse nuclear staining of Nup50, Chat observed “within the nucleus” vs completely cytoplasmic). As a result, the presented quantification appears to be an artifact of this major experimental issue.

All images were taken in the exact same conditions of illumination and exposure. These data are now presented as replication of previous work and have been sent to supplementary material. Our only claim with these experiments is to show decreased NUP50 levels, which is certainly obvious from the images provided.

7. Again, lines 345-364 “mice displayed abnormal Nup50 immunoreactivity characterized by a perinuclear ring”. This is not abnormal. This is in fact where Nups should localize when analyzing a central z plane of a nucleus.

NUP50 is also a chromatin associated NUP, and is not exclusively sitting in the nuclear pore. We however feel this is not a major claim of the manuscript that NUP50 immunoreactivity is morphologically altered in SOD1 mice, and removed this sentence. We now restrict our claims to decreased levels, which is obvious from the images and the quantification.

8. In figure 5e, the Chat and Nup50 staining need to be presented as individual images. As the figure stands now, the Nup50 signal is completely masked by Chat in some images leading the reviewer to believe that presented z plane and/or collapsed z stack is not identical across conditions and thus will also drastically effect quantification.

This is now done in revised Figure S8 (former Figure 5e)

9. The authors should confirm their iPSN IF data by western blot for Nup50 levels given the issues mentioned above.

This request for additional experiments is contradictory with the previous comment stating that these data are not original. Furthermore, the use of bulk western blotting on IPS derived motor neurons will be confounded by the purity achieved in motor neurons culture, that is not 100%, and possible heterogeneity.

10. In figure 5f, C9, TDP and FUS should NOT be grouped together for statistical analyses. These are 3 very different genetic mutations and thus should be treated individually.

This was actually done in previous figure 5f, and in current figure S8 in which mutations are presented individually. We previously only performed a nested t-test on pooled cell lines, and now added a Nested ANOVA on individual cell lines to analyze separately the effects of each mutation. All comparisons between control lines and individual mutant lines showed highly significant differences, thus confirming our previous claim. This novel statistical analysis is presented in the revised Figure S8.

11. Why are the authors using mouse hippocampal neurons to demonstrate human ALS relevance of Nup50 KD? At minimum, mouse cortical neurons should also be employed. However, given that the authors clearly have access to control iPSCs, it is unclear why the experiments were not done in this more biologically relevant system.

We acknowledge that the use of iPSC neurons would be valuable, and are actually currently developing NUP50 KO iPSC. This is however a long process, with >12-18 months required for the generation and characterization of such iPSC based models. Use of cell lines and primary mouse neurons is therefore fast and tractable.

In addition, the reviewer seems to ignore in this comment that we have not restricted our analysis to cell lines, but also showed relevance in two in vivo models, including a vertebrate model. We thus consider that this comment stating that we did not use biologically relevant model system is unfair.

12. Further, in reference to Figure 6, there are a number of concerns. 1. By western blot, the authors show no change in Nup62 and Nup153 levels following Nup50 KD. However, they show “mAb414 aggregates”. Curiously, mAb414 recognizes Nup62 and Nup153. The authors do not report on nuclear mAb414 staining, nor do they show individual channel panels for Fig 6c making it incredibly difficult to reconcile and interpret this data. Based on the preliminary evidence shown, it would be expected that Nup62 and Nup153 levels may be increased.

The statement of the reviewer is at best incomplete: mAb414 is a monoclonal antibody recognizing all nuclear pore proteins carrying an FG repeat, including Nup153 and Nup62, but also (beyond others) Nup107, NUP84, NUP155, or NUP98.

While this includes Nup62 and NUP153, this is far from being restricted to these two proteins. It is thus wrong to reason that the alteration in mAb414 staining should translate directly in alterations in Nup153 and Nup62.

2. The images presented in Fig6c are extremely poor quality. mAb414 staining in the siControl panel looks simply like background staining. There should be a distinct bright rim of immunostaining surrounding DAPI with minimal cytoplasmic staining. Thus, one has to question whether their si-Nup50 staining is simply an artifact.

The images presented in Previous Figure 6C are epifluorescence images, and not confocal or superresolution images, and it is thus absolutely normal that the staining does not show the nuclear rim as suggested by the reviewer. These experiments are not intended (nor designed) to study NPC morphology. Here, we simply illustrate the fact that cytoplasmic NPC-immunoreactive inclusions appear in Nup50 siRNA treated cells, which is very clear from images provided. If the reviewer finds it useful, we can provide alternative images, separate channels or reproduce the experiments with confocal imaging. However, we feel that our claim of mAb414 positive cytoplasmic staining is clearly supported by the current evidence.

In Fig6h, there is quite the range of cell death. In fact, only 1 of the replicates shows increased cell death whereas the

other 2 are nearly identical to controls. Thus, I question the validity of the conclusion that decreased Nup50 increases neuronal death.

We acknowledge that there is heterogeneity in cell death both in HT22 and primary neurons. This is now clearly stated. The reason underlying this heterogeneity is technical: we measure cell death in GFP+ FACS sorted cells, to isolate transfected cells, and transfection efficacy is relatively variable from one experiment to the other.

However, despite heterogeneity, we observe significantly increased cell death in both models, thus invalidating the questioning of the reviewer.

13. The authors broadly assay Ran and RanGAP1 expression but also show defects in the localization of a NCT reporter upon Nup50 KD. Given that the authors highlight the interaction between Nup50 and importins earlier in the text/figures, they should additionally evaluate importin levels, function, and localization in the context of Nup50 KD.

We agree that it is important in future studies to disentangle the consequences of Nup50 loss. As suggested by the reviewer, and shown by the Rothstein group, Ran/RanGAP1 and importin proteins could be involved in ALS-related NPC defects, yet they are not the only ones, and multiple mechanisms could be elicited upon NUP50 loss. Thus, we think that, while this suggestion is interesting, it goes beyond the scope of this current manuscript.

14. The in vivo Nup50 KD studies in both flies and fish are incredibly difficult to interpret and the data presented do not support the authors claims. 1. In both models, the authors chronically KD Nup50. As NPC function and composition is critical for cell division, one can not disentangle developmental vs degenerative effects of reduced Nup50 expression.

The mutations of NUP50 as well as the common NUP50 variant is also present during development. There is thus no reason to exclude a possible developmental contribution of NUP50 altered expression to ALS in patients carrying variants. Such an interesting question goes beyond the current study.

2. In flies and fish, it is critical that the authors show that Nup50 protein was reduced.

There is however no available NUP50 antibody working in flies and fish.

To nevertheless demonstrate the specificity of effect, we actually provide

- in flies: RT-qPCR data showing knock down efficacy (current Figure 6a)

- in fish: rescue of the effect of the morpholino by the NUP50 human cDNA (Figures 6g-k)

These two sets of control experiments are standard controls for fish and flies experiments to circumvent the lack of appropriate antibodies in these two species.

3. In flies, ubiquitous reduction of Nup50, but not motor neuron specific depletion, resulted in motor deficits. This data alone suggests a large non-neuronal contribution of Nup50 KD to behavioral abnormalities. Thus, any NMJ or motor axon phenotypes may be the result of non-cell autonomous defects. The authors begin the manuscript by highlighting that genetic variants are found within genes presumably predominantly expressed in neurons.

The reviewer misunderstood the experiment: this is exactly the opposite. We knocked down NUP50 only in motor neurons, and obtained a motor deficit. All the other cells retained normal NUP50 expression. There is no data in the manuscript studying behavioral consequences of ubiquitous reduction of NUP50. Thus this comment is not valid.

15. It is unclear how the authors performed their statistical analyses. The authors need to explicitly state whether the average of all neurons analyzed from a single patient, mouse, iPSN culture etc was taken as $n = 1$ or whether each individual cell analyzed represents $n = 1$. The latter is inappropriate.

We generally used nested tests (either t-tests or Anova) when we have biological replicates and for each biological replicate multiple technical replicates. This allows to take into account the whole set of information. This is clearly stated in the figure legends, and has been expanded to be clearer in the methods section.

16. The manuscript lacks the critical experiments which would indicate Nup50 variant induced decreased expression is a risk factor for ALS. It should be absolutely required that the authors generate iPSNs with specific Nup50 variants (in an otherwise completely wildtype iPSN) and evaluate Nup50 expression, and multiple ALS phenotypes and pathologies (eg nucleocytoplasmic transport, TDP-43 function and localization, neuronal survival, axon growth etc). Without these experiments, this manuscript simply identifies

Nup50 variants in an ALS population and confirms previous reports of decreased Nup50 expression in familial and sporadic forms of ALS.

The development of custom iPSC models requires 8-12 months, even via experienced providers. The following studies then require 12-18 extra months. Importantly, the generation of NUP50 variants is not doable via standard providers due to difficulties in the genomic site (Synthego, personal communication).

The reviewer is thus asking experiments that, collectively, would delay the publication of this manuscript for probably more than 2 years, and that, for some of them, are not even doable using experienced providers. We thus think this request goes way beyond the current study.

In addition, this reviewer focused only on the NUP50 part of our manuscript, leaving out the genomic and TWAS part, that provides a number of important results to the ALS community.

Minor Concerns:

1. In the results sections for their genetic analyses, it might be helpful if the authors could give a brief less technical description. Much of the technical jargon could be moved to the methods or supplement. This would make the manuscript more accessible to a broad scientific audience.

We agree with the reviewer. We have completely redesigned and rewritten these parts to make it more broadly accessible.

2. The authors comment that ALS risk loci were present in excitatory and inhibitory neurons and to a lesser extent oligodendrocytes (line 131-132). Could they please comment on the identity of these oligodendrocytes? For example, are they mature oligodendrocytes or OPCs? What was the marker used to identify them? Neurons and OPCs derive from the same lineage so the population that is similar to excitatory and inhibitory neurons may in fact be this common precursor cell.

We used single cell ATACseq datasets that identified open chromatin regions in specific cell types. This is not based on the use of markers only, but rather genome wide analysis of cell cluster followed by in-depth identification of cell-types [1]. Clustering analysis was performed using an iterative latent semantic indexing (LSI), which unbiasedly identifies cell-clusters. Then, for each cluster, we manually identified cell types based on the promoter activity and gene activity score in a set of given genes. For instance, oligodendrocytes (Cluster 19-23) were identified based on the promoter activity of the gene MAG and SOX10 while promoter activity in the PDGFRA gene was associated with OPCs (Cluster 12-13) .

3. In their exome-wise association analysis (line 286-287), the authors evaluate nearly 3X more ALS cases compared to controls. Given the genetic heterogeneity amongst the population, this seems like a disproportionately low number of controls especially when compared to earlier analytics when control number matched or exceeded ALS. Can the authors please comment on this?

We indeed used the project MINE database which contains for now ~3X more cases than controls. However, we now added a 2nd cohorts of clinically diagnosed cases (~3000) and controls (~2000) and we observed a significant association after controlling for the 12 constrained genes (Bonferoni $p < 0.05$). Moreover, unbalanced cases-controls analysis is not a limitation since firth bias logistic regression performed in our RVBA analysis prioritizes allele frequencies and controls for unbalanced cases-controls study. However, we agree with the reviewer that having more control individuals increases statistical power but rely on the broad availability of whole genome-sequencing.

4. Line 927 (Fig.)

Reviewer #2 (Remarks to the Author):

The study includes the application of transcriptome-wide association analysis to studying the genetic architecture of ALS. Using TWAS, the authors find 59 genes significantly associated with ALS. Among the newly associated genes (45), the authors observed the decreased expression of NUP50, encoding a nuclear pore complex protein. In follow-up experiments, they show decrease expression of NUP50 in two patients with NUP50 variants. In patient 1 (carrying the NUP50 frameshift Phe58fs mutation), however, decreased protein levels were not associated to decreased mRNA levels (Figure 4 legend). In patient 2 (carrying the near-splice NUP50 mutation c.1086-6C>T), assessment of NUP50 mRNA levels in motor cortex was not quantitative and the RT-PCR shown in Figure 4 (panel e), shows hardly any change in steady state NUP50 mRNA levels.

We agree with the reviewer that the data previously included on the near-splice mutation were not solid enough for publication. We removed mention to this patient and his pathological study that were presented in previous figure 4e and f to avoid premature claims.

The lack of changes in steady state NUP50 mRNA levels in cells with NUP50 variants seem to contradict the main assumption of the TWAS study.

It is important to keep in mind that we obtained two independent evidence for NUP50 involvement. First, the TWAS study shows association of a common NUP50 variant with ALS. Second, and independently, we identified an increased burden of rare variants in NUP50 in ALS and FTD patients. These are two independent genetic mechanisms converging to NUP50 loss. Thus, the two cited piece of evidence are independent and not contradictory. To better explain this point, we added a summary figure detailing the major results of the study.

To address reviewer's comments, we extended largely the part on NUP50 genetics, showed that NUP50 is bound and regulated by TDP43 and FUS. Most importantly, we show that NUP50 mRNA levels are decreased in cortex of ALS patients (new Figure 4G).

Reviewer #3 (Remarks to the Author):

This manuscript from Megat and colleagues is composed by two parts.

In the first part, the latest ALS GWAS is analysed in combination with a) cell-specific ATAC-seq data to define different cell type contributions, b) eQTL, meQTL, sQTL and pQTL data to identify processes relevant to ALS (and not to other NDDs). Results are then complemented by PRS scores, from large DNA resources.

In the second part, to identify novel ALS genes, the group narrows down the investigations to the role of NUP50, a nuclear pore gene that has previously been described to be altered in ALS post mortem brains and iPSC neurons, but has never been genetically linked to ALS. Beyond identifying NUP50 as one of the genes emerging, the authors search for and find possible pathogenic variants linked to ALS, thus linking NUP50 genetics to the disease. Overall this is a vast amount of work.

We thank the reviewer for this positive appreciation. We now provide much better genetic evidence with a second replication cohort in RVBA, and a more straightforward and logical prioritisation of NUP50. To avoid confusion, we removed Figure 2 (PRS).

Generally, it would be great if the authors could expand on their analysis of brain and patient cells in order to build more support for the pathogenicity.

We agree with this comment and provide in this revised version more evidence linking NUP50 to pathology.

First, we added mechanistic insights into relationships between NUP50, TDP43 and FUS (new figure 4D, 4E, 4F)

The drosophila and zebrafish experiments are interesting, but also would benefit from investigations to add support for the partial loss of NUP50 in being pathogenic.

Following are my queries:

- Protein and RNA data from Figure 4 should be harmonised. it would be important to use QPCR on brain samples (4E), as in the previous panel with lymphoblasts.

All three reviewers rightfully questioned the results of Figure 4e and 4f, and in particular the pathogenicity of the splice-site mutation. We agree with the reviewer in that the inclusion of these data was premature and removed all the figures related to this patient (Figure 4e, 4f and S9).

- Pathology in 4F is unclear. Some of the highlighted changes appear to be present in control as well. Quantification of features such as the ring-shaped perinuclear staining is needed in order to be convincing.

We agree with the reviewer in that these data would require much deeper analysis and removed all the figures related to this patient (Figure 4e, 4f and S9).

- The fact that NUP50 decreases also in control ALS is interesting and in line with previous other findings, but it does not help convince the reader about the pathogenicity of the mutation. Could the authors perform blots in other unaffected regions (eg cerebellum)? One would expect a similar decrease in the case with splicing changes, but not in the ALS control.

This figure has been removed to avoid premature claims.

- Similarly, to add support to the pathogenicity claims, as Coyne et al have shown reduction in other proteins in their extensive nuclear pore characterisation, it would be very useful if investigations in the lymphoblasts and brains were extended to at least one that changes and one that seems to remain unchanged.

We agree with the reviewer in that these data would require much deeper analysis and removed all the figures related to this patient (Figure 4e, 4f and S9).

- In regards to the possible splicing mutant. In Supplementary figure 9 it would be reassuring to see detection of the SNPs in equal amounts using genomic DNA, though I appreciate the exact same combination of primers may not be used as these are for cDNA.

We agree with the reviewer in that these data would require much deeper analysis and removed all the figures related to this patient (Figure 4e, 4f and S9).

- In figure 4F, which is the ALS1 case from 4E?
Figure 4f has been removed.

- In figure 5. Intensity and pathology should be analysed at both early and symptomatic timepoints.

Another reviewer noted that these results were confirmatory of other's work, and we agree with this. We thus removed these data from the main manuscript and provide them as confirmatory evidence in Figure S8.

- Is the intensity in other cell types, eg in dorsal SC unchanged? This would help convince of the specificity.

- Similarly to above, inclusion of other nuclear pore proteins would be useful, to show whether this is specific to NUP50.

NUP50 intensity appears heterogenous across cell types which render answering this question difficult and would require multiple double immunostainings and quantifications. We do not claim for specificity, and now use these results as confirmation of work of others. Data are provided now as supplementary figure 8.

- Results should be also reported on a per animal basis, not just per cell.

We agree, and this is now provided. Please note that this was only a matter of presentation, and had no influence on statistical analysis as we are using nested t-test that takes into account biological replicates (here animals) and technical replicates (here cells).

- Figure 5. Very small error bars in 5F. Can results be shown with scatter plots, and can the different replicates be highlighted by colour?

We now provide results as violin plots, as scatter plots are difficult to read with so many points. Individual replicates are individually very similar and will be provided in source data.

Is this result in line with Coyne et al at earlier timepoints?

As for Coyne et al, we observe progressive loss of NUP50. At 28 days post differentiation, there was not yet a significant loss of NUP50 (see figure below). This is entirely replicative of Coyne et al.

Figure: Quantification of NUP50 levels at D28 shows no significant decrease in fALS mutation carriers

If cultures are pure enough, could authors provide RNA analysis as well?

Unfortunately, the purity of motor neuron culture does not allow relevant bulk RNA experiments.

- The progressive phenotype in flies is interesting. Is there evidence of this being specific for motoneurons? Are there any pathological features that can be investigated beyond just the motor phenotype?

The driver used in Figure 7b-e of the previous manuscript (now Figure 6b-e) is restricted to motor neurons. Hence the phenotype is specific to motor neurons. We also show that post-synaptic apparatus of the neuromuscular junction is affected in Figure 7d-e, likely as a result of a miscommunication between motor neurons, deficient in Nup50, and muscle, still expressing it.

Reference:

- 1 Corces MR, Shcherbina A, Kundu S, Gloudemans MJ, Fresard L, Granja JM, Louie BH, Eulalio T, Shams S, Bagdatli ST et al (2020) Single-cell epigenomic analyses implicate candidate causal variants at inherited risk loci for Alzheimer's and Parkinson's diseases. *Nat Genet* 52: 1158-1168 Doi 10.1038/s41588-020-00721-x

REVIEWER COMMENTS

Reviewer #1 (Remarks to the Author):

The authors have substantially improved the current manuscript by removing the premature cell biology experiments regarding Nup50 and instead focusing on their genetic analyses. While the reviewer strongly agrees with this decision, the manuscript would be of much higher impact and broader interest had the authors taken the time to carry out the proper cell biology experiments to lend essential support to their genetic findings. However, the authors have done a nice job at editing the text to be more accessible to a broader audience. Specific comments on the revised manuscript are detailed below.

Introduction line 97-98, the authors state that nuclear pore dysfunction occurs downstream of TDP-43 and FUS aggregation. That is actually not the case in authentic human neurons. The referenced studies highlight aggregation of cytoplasmic pools of Nups specifically and provide minimal insights into Nups within the NPC itself. In fact, a recent study (Coyne et al 2021 Science Translational Medicine) has shown that reduction of Nups from the NPC occurs well prior to TDP-43 dysfunction and mislocalization in a larger number of sALS iPSCs. Also human biopsy and post mortem studies clearly document that nuclear clearing occurs prior to later nuclear TDP43 aggregates (Seely et al, Brain) Thus the authors should clarify these points within the text.

Figure 4e-g is a bit confusing as described in the text. The authors demonstrated early in their manuscript that Nup50 mRNA levels are unchanged at least in n = 1 patient (Figure 3f). Now the authors show a decrease in Nup50 mRNA in ALS patient cortex. It would be helpful if the authors could comment on the status of Nup50 variants in ALS patients and C9, FUS, and TDP iPSC lines utilized throughout Figure 4. The reviewer questions whether these findings in patient cortex and iPSC lines are simply a reproduction of recent literature (Coyne et al 2020 Neuron and Coyne et al 2021 Science Translational Medicine) or if perhaps there is a more novel biological effect of Nup50 variants.

The manuscript would also benefit from an enhanced discussion/explanation of the findings that Nup50 mRNA expression may be regulated by TDP-43 and/or FUS in a cell type specific manner. It has been recently demonstrated that TDP-43 dysfunction is downstream of Nup alterations in sALS (Coyne et al 2021 Science Translational Medicine). Further, the authors appear to be claiming that genetic variants in Nup50 contribute to ALS. It is widely accepted that genetic alterations are an initiating event in disease and thus it becomes confusing when the authors demonstrate that a later event in ALS (TDP-43 and FUS dysfunction/mislocalization) triggers a similar result to a genetic alteration. Thus, as indicated above, the manuscript would greatly benefit from a targeted analysis of Nup50 variants in the samples used for these studies. Additionally, it might help to clarify some of these points if the authors framed the text in a way that perhaps these studies are verifying a more global role of Nup50 disruption in ALS and not necessarily related to their genetic findings (case in which this becomes 2 distinct stories). It might also help to refer to the recent UNC13A findings.

Reviewer #3 (Remarks to the Author):

The authors have made substantial changes to the manuscript in both its parts.

They have removed the results that were not mature enough, and have added novel analyses and experiments.

Overall this version is greatly improved.

One comment to help the discussion: I find it confusing for the reader to pull together the findings on

Nup50 reduced expression in selective knock-downs and in post mortem brains – including the fact that the clearest consequences appear in FUS knock-down which should not play a role in general ALS cohorts. I would encourage to add a sentence or two in the discussion to address these findings.

Reviewer #4 (Remarks to the Author):

The authors set out to characterize Amyotrophic lateral sclerosis (ALS) heritability and survey molecular mechanisms underlying the phenotype of interest.

A complex study is conducted using different types of information, ranging across GWAS summary statistics, molecular traits, cell type-specific experiments, whole-genome data, biological models, and more.

The authors discuss pertinent questions such as differentiation of splicing effects in ALS compared to other neurodegenerative diseases and effects from other molecular traits.

Unfortunately I find the authors' presentation unclear in general; and in particular, the TWAS-related integrative analysis presents conceptual errors and is superficial.

I appreciate the ambition displayed in this study and the scope of biological information chosen, but I have major concerns with the current state of the manuscript.

Although I think the entire manuscript needs a revision, I'll restrict myself to Statistical Genetics/Integrative Analyses themes to complement the other reviewers' expertise.

Here follow my concerns (major and minor, each tier loosely ordered by appearance in the text):

Major concern 1: Using a single tissue for TWAS is too limited.

Molecular studies typically lack the relevant biological context from specific traits, specially in degenerative diseases.

The dorso-lateral prefrontal cortex study, using samples selected solely on criteria of being subject to Schizophrenia or not, is likely to manifest this lack.

Given the widespread sharing of expression patterns across tissues (established in multiple publications like The GTEx Consortium 2022, <https://www.science.org/doi/full/10.1126/science.aaz1776>), I strongly recommend using more tissues (for example all Brains in the latest GTEx release). This allows to sidestep limitations from inadequate tissue etiology coverage in molecular studies.

It's standard practice when using multiple tissue, to gather all gene-tissue-trait tuples' associations (assumed to be N) and prioritize candidates using Bonferroni correction via N on them.

Another widespread alternative is factoring tissue correlation when conducting multi-tissue associations.

e.g.1: UTMOST using multiple Brains on ASD: Rodriguez-Fontenla, 2021,

<https://www.nature.com/articles/s41398-021-01378-8> (although this publication uses the dated release v6 of GTEx, I recommend v8).

e.g.2: S-MultiXcan using multiple brains on Alzheimer: Gerring et al, 2020,

<https://link.springer.com/article/10.1186/s13195-020-00611-8>.

Major Concern 2: I find what the authors describe as "replication with Hippocampus tissue" to be unacceptable.

First: I interpret the authors use the same underlying GWAS data set and only change "the reference panel" to use Hippocampus data.

The prediction models merely infer molecular traits in a GWAS data set, so I can't accept using a same GWAS data set both for discovery and replication as independent evidence.

Second: the Hippocampus samples are extracted from the same individuals as the dorsolateral brain samples. Given widespread sharing of regulatory mechanisms (The GTEx Consortium, 2020, <https://www.science.org/doi/full/10.1126/science.aaz1776>), prediction in Hippocampus and DLPFC is correlated.

To address replication questions: the authors should use a different approach like analyzing the DLPFC associations in another ALS GWAS study with non-overlapping individuals.

They can use the additional tissue prediction models to other ends like addressing "Major concern 4" or pooling both tissues together (e.g. UTMOST, S-MultiXcan) to get extra power; but definitely not for replication.

Major Concern 3: TWAS only yields associations and doesn't address causality.

For example, TWAS is vulnerable to pleiotropy - or different causal mechanisms in LD: e.g. one variant affecting molecular expression, another affecting the trait, and both in LD.

This is acknowledged by FUSION's authors here: Mancuso et al, 2019

<https://www.nature.com/articles/s41588-019-0367-1>.

Additionally, LASSO models are known to be specially vulnerable to variants in LD (e.g. Ngundu et al

<https://www.sciencedirect.com/science/article/pii/S0002929720300033>).

To address causality: When integrating complex traits with molecular data, it is standard practice to complement associations with other estimates that evaluate causality, such as colocalization methods. The authors don't present any colocalization analyses on the main section. They merely mention they ran COLOC in the supplement, totally disconnected from the rest of the manuscript, without any explanation, justification, context, or analysis of the result.

Furthermore, although COLOC was a seminal method in "complex trait - to - molecular traits" integrative analysis, it is severely limited in its assumption of a single causal variant. COLOC has been effectively superseded by newer developments that support multiple causal variants like eCAVIAR (Hormozdiari et al, <https://www.sciencedirect.com/science/article/pii/S0002929716304396>) or ENLOC (Wen et al <https://journals.plos.org/plosgenetics/article?id=10.1371/journal.pgen.1006646>).

Furthermore: I object to the criteria mentioned in the supplement to define colocalization, based on pooling coloc's H3 and H4 estimates. This definition is neither explained nor justified. H3 is a measure of heterogeneity, of molecular and complex traits having distinct causal variants - basically antipodal to the interest for causality.

In the context of the toolset chosen by the authors, I have two alternatives to recommend:

- if the authors want to prioritize variant-based analyses, given they use CAVIAR, the most natural complement is to use eCAVIAR to address this.

- if they want to focus on gene-based analyses, given the authors' choice for FUSION software, the FOCUS method (Mancuso et al, 2019, <https://www.nature.com/articles/s41588-019-0367-1>) is a good candidate to address causality (e.g. Gerring et al, 2020, <https://link.springer.com/article/10.1186/s13195-020-00611-8>). FOCUS even goes so far as to claim it's robust respect missing measurements in actual causal tissues (in other words identifying relevant mechanisms regardless of actual tissue used).

To make prediction more robust and incorporate causality: It is also common practice to use models informed via fine-mapping to reduce LD pollution and incorporate biological evidence of causality (first presented here <https://www.ncbi.nlm.nih.gov/pmc/articles/PMC7693040/>)

Minor Concern 4: For the "Cell type specific and molecular trait heritability of ALS" section: related to "splicing QTLs were enriched in ALS heritability but not other neurodegenerative diseases": I concur with the authors' interest on this claim, but I consider the follow-up analyses based on annotations (MaxCPP-ExAC) insufficient.

To make this meaningful, I would compare to enrichment from other potentially related (or even unrelated) mechanisms as baselines.

Conceptual example 1: how does the MaxCPP-ExAC enrichment compare to enrichment from an analogous annotation based on RBP (a hypothetical MaxCPP-RBP annotation)?

Conceptual example 2: Other studies establish that the Major Histocompatibility Complex plays a role in ALS. How does MaxCPP-ExAC enrichment compare to MHC enrichment?

Minor concern 5: Line 190: "[...] (Fig. 1c, Table S3). Suggesting [...]" sic: the syntax here is confusing. Was it supposed to be one compound sentence?

Minor concern 6: Line 283: "Non-overlapping with Project Mine": this statement is missing context. It's the first reference to Project Mine in this manuscript.

I'm not familiar with Project Mine, and can't immediately infer from reading the previous sections what scope is sufficient for the "non-overlapping with Project Mine" requirement.

I had to actually skim through other Project Mine publications to try to guess at this, and which specific parts of the study is entirely contained within Project Mine data.

I think a brief summary in the introduction of Project Mine could go a long way to make this part easier to understand.

Minor concern 7: Table 1: Please add chromosome and position to this table. It's very useful to visualize how the implicated genes distribute across the genes.

Minor concern 8: Discussion: line 551: "genes associated to mitochondrial physiology" is stated but neither do the authors explain this nor include a reference.

Sincerely,

Alvaro Barbeira

Comments from reviewers are in black
Our answers in red

All changes made to the manuscript were highlighted in yellow in the revised file.

Reviewer #1 (Remarks to the Author):

The authors have substantially improved the current manuscript by removing the premature cell biology experiments regarding Nup50 and instead focusing on their genetic analyses. While the reviewer strongly agrees with this decision, the manuscript would be of much higher impact and broader interest had the authors taken the time to carry out the proper cell biology experiments to lend essential support to their genetic findings. However, the authors have done a nice job at editing the text to be more accessible to a broader audience. Specific comments on the revised manuscript are detailed below.

We thank the reviewer for this positive appreciation of our revision. We agree that additional cell biology experiments are important for further extending genetic findings, and would respectfully argue that the generation and characterization of iPS derived models is a long standing task that goes beyond the current manuscript.

Introduction line 97-98, the authors state that nuclear pore dysfunction occurs downstream of TDP-43 and FUS aggregation. That is actually not the case in authentic human neurons. The referenced studies highlight aggregation of cytoplasmic pools of Nups specifically and provide minimal insights into Nups within the NPC itself. In fact, a recent study (Coyne et al 2021 Science Translational Medicine) has shown that reduction of Nups from the NPC occurs well prior to TDP-43 dysfunction and mislocalization in a larger number of sALS iPSCs. Also human biopsy and post mortem studies clearly document that nuclear clearing occurs prior to later nuclear TDP43 aggregates (Seely et al, Brain) Thus the authors should clarify these points within the text.

We acknowledge that our previous formulation was missing the possibility that dysfunction of the nuclear pore could also operate before TDP43 or FUS aggregation. To answer the reviewer's comment and present a balanced review of the current literature, we reworded these two sentences and state that both events reciprocally influence each other.

This is now worded as such: lines 98-99 of the revised manuscript:

"Nuclear pore dysfunction precedes TDP-43 mislocalisation in neurons of sporadic ALS patients, ⁴ and reciprocally is exacerbated by TDP-43 and FUS aggregation ^{5,6}."

Importantly, this is also largely acknowledged in the discussion section (see below).

Figure 4e-g is a bit confusing as described in the text. The authors demonstrated early in their manuscript that Nup50 mRNA levels are unchanged at least in n = 1 patient (Figure 3f). Now the authors show a decrease in Nup50 mRNA in ALS patient cortex. It would be helpful if the authors could comment on the status of Nup50 variants in ALS patients and C9, FUS, and TDP iPSC lines utilized throughout Figure 4. The reviewer questions whether these findings in patient cortex and iPSC lines are simply a reproduction of recent literature (Coyne et al 2020 Neuron and Coyne et al 2021 Science Translational Medicine) or if perhaps there is a more novel biological effect of Nup50 variants.

We thank the reviewer for this comment. There is no ALS patients iPSC cell line used in Figure 4. In figure 4e-g, we use publicly available datasets of either K562 (4e), SH5SY (4f) or wild type iPSC neurons (4f) with knock down of the considered RBPs. Thus, the NUP50 status is the same in all cell lines and cannot confound the result.

The referee is probably referencing to figure S8 of the previous manuscript (S3 of the current version). In this supplementary figure, we agree with the reviewer that we are reproducing results from Pr Rothstein laboratory, and do not claim for any strong novelty here. We however think it is valuable to include these panels, and stress in the revised manuscript that this is consistent with previous results. These sentence now reads:

"In addition, we observed reduced expression of NUP50 in hiPSC-derived motoneurons of patients with ALS mutations, including *FUS* (n=2), *TARDBP* (n=2), and *C9ORF72* (n=2) (Fig S3a-b) as well

as in motor neurons of mice expressing either ALS linked mutant SOD1 or mutant FUS (Fig S3c-d), consistent with previously published results^{4,7}.”

The manuscript would also benefit from an enhanced discussion/explanation of the findings that Nup50 mRNA expression may be regulated by TDP-43 and/or FUS in a cell type specific manner. It has been recently demonstrated that TDP-43 dysfunction is downstream of Nup alterations in sALS (Coyne et al 2021 Science Translational Medicine). Further, the authors appear to be claiming that genetic variants in Nup50 contribute to ALS. It is widely accepted that genetic alterations are an initiating event in disease and thus it becomes confusing when the authors demonstrate that a later event in ALS (TDP-43 and FUS dysfunction/mislocalization) triggers a similar result to a genetic alteration. Thus, as indicated above, the manuscript would greatly benefit from a targeted analysis of Nup50 variants in the samples used for these studies. Additionally, it might help to clarify some of these points if the authors framed the text in a way that perhaps these studies are verifying a more global role of Nup50 disruption in ALS and not necessarily related to their genetic findings (case in which this becomes 2 distinct stories). It might also help to refer to the recent UNC13A findings.

We agree with the reviewer that our manuscript suggests a dual relationship between NUP50 and TDP43/FUS: TDP-43 and FUS regulates NUP50 mRNA levels, while genetically predetermined lower levels of NUP50 in the CNS might predispose to TDP43 aggregation as suggested by Coyne and collaborators. We agree that this dual relationship might not have been clear in the previous version of the manuscript.

To answer the reviewer’s comment, we reorganized the discussion section, emphasizing separately on (i) loss of NUP50 and its consequences (in particular on TDP43), and (ii) regulation of NUP50 by TDP-43 and FUS

We also, as requested by the reviewer, more directly refer to the Coyne et al studies to clarify our discussion.

The whole paragraph discussing the role of NUP50 in ALS has been rewritten as such:

“Our study provides two additional genetic evidence linking NUP50 and ALS. First, a common variant causing decreased expression of *NUP50* is associated to ALS. Second, rare variants are enriched in *NUP50* and, at least some of them lead to loss of the protein (Fig 7.).

What could be the consequences of *NUP50* loss in ALS? *Nup50* knock out mice die *in utero* due to severe neural tube defects⁸ suggesting a critical function of NUP50 in the central nervous system. Consistently, *Nup50* knockdown in primary neurons or in HT22 cell lines increased neuronal death, and knock-down of *Nup50* in *Drosophila* motor neurons led to a mild loss of motor function accompanied by decreased neuromuscular junction size. This is in agreement with previous studies showing normal motor neuron development in *Drosophila*⁵ upon *Nup50* loss, as the defect appeared later in life, consistent with ALS adult onset. Similarly, *nup50* knockdown in zebrafish led to reduction of evoked swimming bouts, without any apparent developmental deficits, and was associated with abnormal motor neurons. Thus, loss of function of NUP50 is toxic to motor neurons, and sufficient to lead to motor neuron disease.

Loss of NUP50 could lead to multiple cellular consequences, that could each contribute to motor neuron demise. Indeed, *Nup50* knockdown triggered cytoplasmic inclusions of nuclear pore components, as well as of RanGAP1, a key protein regulating nucleocytoplasmic shuttling function, as observed in ALS patients^{5,9}, and impaired nuclear pore function. Importantly, Coyne et al recently showed that nuclear pore alterations were sufficient to lead to TDP-43 dysfunction in human neurons and ALS patients^{4,7}. Thus, lower levels of NUP50 might by themselves contribute to nuclear pore dysfunction, and, in turn, to TDP-43 dysfunction. In our cell models however, acute *Nup50* knockdown was not sufficient to trigger aggregation or mislocalization of endogenous TDP-43. Our results also suggest that NUP50 levels could conversely be modulated by TDP-43 or FUS dysfunction. Indeed, *NUP50* mRNA binds TDP-43 and FUS, and its levels are regulated by them. Consistently, *NUP50* is less expressed in ALS cortex and multiple models of ALS⁷. It is thus tempting to speculate that TDP-43 or FUS dysfunction could further enhance NUP50 loss in ALS patients, in a vicious pathogenic cycle. More research is needed to characterize the pathogenic relationships between NUP50 and TDP 43 or FUS aggregation and dysfunction in ALS.”

Of note, we respectfully disagree with the statement of the reviewer that characterizing NUP50 common variant genotype in the studied samples would provide a pathogenic link. Demonstrating such causality would require also to modify NUP50 locus in various iPSC models (either with familial mutations or sporadic) to determine the individual net effect of NUP50 genotype. These are ongoing efforts in our lab, that will require >2 years to reach a definitive conclusion, and we thus think that such experiments go beyond the current manuscript.

The UNC13A findings are referenced as ref. 13 and 14 of the manuscript, in the introduction and in the discussion.

Reviewer #3 (Remarks to the Author):

The authors have made substantial changes to the manuscript in both its parts.

They have removed the results that were not mature enough, and have added novel analyses and experiments.

Overall this version is greatly improved.

One comment to help the discussion: I find it confusing for the reader to pull together the findings on Nup50 reduced expression in selective knock-downs and in post mortem brains – including the fact that the clearest consequences appear in FUS knock-down which should not play a role in general ALS cohorts. I would encourage to add a sentence or two in the discussion to address these findings.

We thank the reviewer for this comment. We rewrote this part of the discussion to better articulate our reasoning.

Reviewer #4 (Remarks to the Author):

The authors set out to characterize Amyotrophic lateral sclerosis (ALS) heritability and survey molecular mechanisms underlying the phenotype of interest.

A complex study is conducted using different types of information, ranging across GWAS summary statistics, molecular traits, cell type-specific experiments, whole-genome data, biological models, and more.

The authors discuss pertinent questions such as differentiation of splicing effects in ALS compared to other neurodegenerative diseases and effects from other molecular traits.

Unfortunately I find the authors' presentation unclear in general; and in particular, the TWAS-related integrative analysis presents conceptual errors and is superficial.

I appreciate the ambition displayed in this study and the scope of biological information chosen, but I have major concerns with the current state of the manuscript.

Although I think the entire manuscript needs a revision, I'll restrict myself to Statistical Genetics/Integrative Analyses themes to complement the other reviewers' expertise.

Further to reviewer's comment, the whole manuscript was reformatted. We removed 7 supplementary figures, streamlined the results and rewrote entirely the discussion on NUP50 role in ALS. We hope that these major changes will convince the reviewer in addition to the additional analyses performed in response to his comments below.

Here follow my concerns (major and minor, each tier loosely ordered by appearance in the text):

Major concern 1: Using a single tissue for TWAS is too limited.

Molecular studies typically lack the relevant biological context from specific traits, specially in degenerative diseases.

The dorso-lateral prefrontal cortex study, using samples selected solely on criteria of being subject to Schizophrenia or not, is likely to manifest this lack.

Given the widespread sharing of expression patterns across tissues (established in multiple

publications like The GTEx Consortium

2022, <https://www.science.org/doi/full/10.1126/science.aaz1776>), I strongly recommend using more tissues (for example all Brains in the latest GTEx release). This allows to sidestep limitations from inadequate tissue etiology coverage in molecular studies.

It's standard practice when using multiple tissue, to gather all gene-tissue-trait tuples' associations (assumed to be N) and prioritize candidates using Bonferroni correction via N on them.

Another widespread alternative is factoring tissue correlation when conducting multi-tissue associations.

e.g.1: UTMOST using multiple Brains on ASD: Rodriguez-Fontenla,

2021, <https://www.nature.com/articles/s41398-021-01378-8> (although this publication uses the dated release v6 of GTEx, I recommend v8).

e.g.2: S-MultiXcan using multiple brains on Alzheimer: Gerring et al,

2020, <https://link.springer.com/article/10.1186/s13195-020-00611-8>.

We thank the reviewer for this comment and we agree that using multiple tissues reduces bias coverage from individual study.

Therefore, as suggested by the reviewer, we performed TWAS analysis on all brains tissue from the GTEx consortium v8 (n=13 tissues). As requested, we performed Bonferroni correction on N tissues to prioritize gene-trait associations. The results for the discovery and replication cohort can be found in the updated Figure 2 and Table 1.

Major Concern 2: I find what the authors describe as "replication with Hippocampus tissue" to be unacceptable.

First: I interpret the authors use the same underlying GWAS data set and only change "the reference panel" to use Hippocampus data.

The prediction models merely infer molecular traits in a GWAS data set, so I can't accept using a same GWAS data set both for discovery and replication as independent evidence.

Second: the Hippocampus samples are extracted from the same individuals as the dorsolateral brain samples. Given widespread sharing of regulatory mechanisms (The GTEx Consortium, 2020, <https://www.science.org/doi/full/10.1126/science.aaz1776>), prediction in Hippocampus and DLPFC is correlated.

To address replication questions: the authors should use a different approach like analyzing the DLPFC associations in another ALS GWAS study with non-overlapping individuals.

They can use the additional tissue prediction models to other ends like addressing "Major concern 4" or pooling both tissues together (e.g. UTMOST, S-MultiXcan) to get extra power; but definitely not for replication.

We thank the reviewer for this comment, and we also agree that using the same underlying GWAS and switching the reference panel shouldn't have been termed "replication" in the previous manuscript. It is certainly true that an ideal replication would include a similar size cohort for GWAS as in the discovery cohort. However, ALS is a rare disease, and there are few large size cohorts with available genotype. We also provide independent validation of NUP50 loss through rare variant burden analysis, that is replicated in two independent WGS cohorts.

To address the reviewer comment, we decided to perform a 2 step TWAS discovery and replication with non-overlapping individuals.

The discovery cohort consisted in 12,577 cases and 23,475 controls from the previous ALS GWAS¹

To assemble the replication cohort, we included ALS cases non-overlapping with the discovery (Supplementary methods) which after QC yielded 8,214 cases and 14,129 controls². We also aggregated ALS cases from a recently described cohort of ALS patients which yielded after QC 1,821 ALS cases and 2,010 controls³. A generalized linear mixed models was fitted using SAIGE per cohort and both cohorts were merged using an inverse variance method on a total of 10,035 cases and 16,139 controls.

The description and results of the discovery and replication can be found in Figure 2, Table 1. The association analysis and summary statistics of the replication GWAS will be deposited in a public repository upon publication.

We observed 3 genes reaching genome-wide significant among them 2 known ALS loci and NUP50 (Figure 2, Table 1). We then performed the replication TWAS, that replicated the 3 previous associations. Most importantly, we meta-analyzed both cohorts with a weighted Z-score using the Stouffer' method, which yielded 3 novel TWAS association among them 1 known ALS gene. All updated datasets can be found in Figure 2 and Table 1. Description of the cohorts and association analysis are described in Supplementary methods.

Major Concern 3: TWAS only yields associations and doesn't address causality.

For example, TWAS is vulnerable to pleiotropy - or different causal mechanisms in LD: e.g. one variant affecting molecular expression, another affecting the trait, and both in LD.

This is acknowledged by FUSION's authors here: Mancuso et al, 2019 <https://www.nature.com/articles/s41588-019-0367-1>.

Additionally, LASSO models are known to be specially vulnerable to variants in LD (e.g. Ngundu et al <https://www.sciencedirect.com/science/article/pii/S0002929720300033>).

To address causality: When integrating complex traits with molecular data, it is standard practice to complement associations with other estimates that evaluate causality, such as colocalization methods.

The authors don't present any colocalization analyses on the main section. They merely mention they ran COLOC in the supplement, totally disconnected from the rest of the manuscript, without any explanation, justification, context, or analysis of the result.

Furthermore, although COLOC was a seminal method in "complex trait - to - molecular traits" integrative analysis, it is severely limited in its assumption of a single causal variant. COLOC has been effectively superseded by newer developments that support multiple causal variants like eCAVIAR (Hormozdiari et al, <https://www.sciencedirect.com/science/article/pii/S0002929716304396>) or ENLOC (Wen et al <https://journals.plos.org/plosgenetics/article?id=10.1371/journal.pgen.1006646>). Furthermore: I object to the criteria mentioned in the supplement to define colocalization, based on pooling coloc's H3 and H4 estimates. This definition is neither explained nor justified. H3 is a measure of heterogeneity, of molecular and complex traits having distinct causal variants - basically antipodal to the interest for causality.

In the context of the toolset chosen by the authors, I have two alternatives to recommend:

- if the authors want to prioritize variant-based analyses, given they use CAVIAR, the most natural complement is to use eCAVIAR to address this.

- if they want to focus on gene-based analyses, given the authors' choice for FUSION software, the FOCUS method (Mancuso et al, 2019, <https://www.nature.com/articles/s41588-019-0367-1>) is a good candidate to address causality (e.g. Gerring et al, 2020, <https://link.springer.com/article/10.1186/s13195-020-00611-8>). FOCUS even goes so far as to claim it's robust respect missing measurements in actual causal tissues (in other words identifying relevant mechanisms regardless of actual tissue used).

To make prediction more robust and incorporate causality: It is also common practice to used models informed via fine-mapping to reduce LD pollution and incorporate biological evidence of causality (first presented here <https://www.ncbi.nlm.nih.gov/pmc/articles/PMC7693040/>)

We agree with the reviewer that TWAS does not address causality and is especially vulnerable to pleiotropy, co-regulation and/or finite size reference panel as it has been described recently¹⁰.

Therefore, as suggested by the reviewer we choose to perform gene-based analyses using FOCUS to finemap TWAS associations. We ran FOCUS for each locus that reached genome-wide significant in discovery and replication cohorts. Our results suggest that C9orf72 is the causal gene on locus 9:26112447-9:28222934, which is supported by the known role of C9orf72 intronic expansion in ALS. Moreover, we observed a posterior probability of 0.82 on the locus 22: 44996472-22:46463431, suggesting that NUP50 is likely the causal gene in this region. This is now included in Table 1.

Minor Concern 4: For the "Cell type specific and molecular trait heritability of ALS" section:

related to "splicing QTLs were enriched in ALS heritability but not other neurodegenerative diseases":

I concur with the authors' interest on this claim, but I consider the follow-up analyses based on annotations (MaxCPP-ExAC) insufficient.

To make this meaningful, I would compare to enrichment from other potentially related (or even unrelated) mechanisms as baselines.

Conceptual example 1: how does the MaxCPP-ExAC enrichment compare to enrichment from an analogous annotation based on RBP (a hypothetical MaxCPP-RBP annotation)?

Conceptual example 2: Other studies establish that the Major Histocompatibility Complex plays a role in ALS. How does MaxCPP-ExAC enrichment compare to MHC enrichment?

We thank the reviewer for this comment and we agree that the results of MaxCPP-ExAC annotation should be considered in the light of other related mechanism as baseline. We indeed have performed such analysis in Figure 2b as described by Hormozdiari et al ¹¹

We performed conditional LD score regression while accounting for 72 baseline models as well as other Max-CPP annotation in novel Figure 2b. We decided to focus on mRNA-related mechanisms and therefore removed protein QTLs data as we believed it is beyond the scope of this study.

The novel Figure 2b now shows conditional enrichment of each annotation individually compared to the other Max-CPP annotation and the baseline model. We observed that splicing QTLs are enriched in ALS and Parkinson's disease as well as in Schizophrenia. Moreover, splicing QTLs-ExAC annotation is enriched in Parkinson's disease and Schizophrenia while only a trend is observed for ALS. All updated data are shown in Figure 2b.

Finally, as we do think that creating a Max-CPP-RBP annotation would be meaningful it is conceptually difficult as the Max-CPP RBP is a positional-based annotation which depends on whether a specific variant overlap or do not overlap a given RBP binding site. To create, such Max-CPP annotation variant effect sizes on each molecular phenotypes (expression, splicing, histones) is needed in order to establish causal posterior probabilities for each variant accounted for LD between variants. Therefore, it seems difficult to create such annotation for RBP binding sites.

Minor concern 5: Line 190: "[...] (Fig. 1c, Table S3). Suggesting [...]" sic: the syntax here is confusing. Was it supposed to be one compound sentence?

The mentioned sentence has been deleted as it refers to previous Figure 1c, that we removed further to the previous comment.

Minor concern 6: Line 283: "Non-overlapping with Project Mine": this statement is missing context. It's the first reference to Project Mine in this manuscript.

I'm not familiar with Project Mine, and can't immediately infer from reading the previous sections what scope is sufficient for the "non-overlapping with Project Mine" requirement.

I had to actually skim through other Project Mine publications to try to guess at this, and which specific parts of the study is entirely contained within Project Mine data.

I think a brief summary in the introduction of Project Mine could go a long way to make this part easier to understand.

We agrees with the reviewer and we think that mentioning Project mine dataset in the text is confusing. We modified the text in the RVBA analysis and rename the dataset as discovery and replication cohorts.

Minor concern 7: Table 1: Please add chromosome and position to this table. It's very useful to visualize how the implicated genes distribute across the genes.

This has been done.

Minor concern 8: Discussion: line 551: "genes associated to mitochondrial physiology" is stated but neither do the authors explain this nor include a reference.

The very stringent analysis requested by this reviewer in major comments 1 and 2 restricted widely the list of TWAS significant genes, including most mitochondrially related genes. We thus removed these sentences from the discussion.

References

- 1 van Rheenen, W. *et al.* Genome-wide association analyses identify new risk variants and the genetic architecture of amyotrophic lateral sclerosis. *Nat Genet* **48**, 1043-1048, doi:10.1038/ng.3622 (2016).
- 2 Nicolas, A. *et al.* Genome-wide Analyses Identify KIF5A as a Novel ALS Gene. *Neuron* **97**, 1268-1283 e1266, doi:10.1016/j.neuron.2018.02.027 (2018).
- 3 Grassano, M. *et al.* Systematic evaluation of genetic mutations in ALS: a population-based study. *J Neurol Neurosurg Psychiatry*, doi:10.1136/jnnp-2022-328931 (2022).
- 4 Coyne, A. N. *et al.* Nuclear accumulation of CHMP7 initiates nuclear pore complex injury and subsequent TDP-43 dysfunction in sporadic and familial ALS. *Sci Transl Med* **13**, doi:10.1126/scitranslmed.abe1923 (2021).
- 5 Chou, C. C. *et al.* TDP-43 pathology disrupts nuclear pore complexes and nucleocytoplasmic transport in ALS/FTD. *Nat Neurosci* **21**, 228-239, doi:10.1038/s41593-017-0047-3 (2018).
- 6 Lin, Y. C. *et al.* Interactions between ALS-linked FUS and nucleoporins are associated with defects in the nucleocytoplasmic transport pathway. *Nat Neurosci*, doi:10.1038/s41593-021-00859-9 (2021).
- 7 Coyne, A. N. *et al.* G4C2 Repeat RNA Initiates a POM121-Mediated Reduction in Specific Nucleoporins in C9orf72 ALS/FTD. *Neuron* **107**, 1124-1140 e1111, doi:10.1016/j.neuron.2020.06.027 (2020).
- 8 Smitherman, M., Lee, K., Swanger, J., Kapur, R. & Clurman, B. E. Characterization and targeted disruption of murine Nup50, a p27(Kip1)-interacting component of the nuclear pore complex. *Mol Cell Biol* **20**, 5631-5642, doi:10.1128/MCB.20.15.5631-5642.2000 (2000).
- 9 Zhang, K. *et al.* The C9orf72 repeat expansion disrupts nucleocytoplasmic transport. *Nature* **525**, 56-61, doi:10.1038/nature14973 (2015).
- 10 Wainberg, M. *et al.* Opportunities and challenges for transcriptome-wide association studies. *Nat Genet* **51**, 592-599, doi:10.1038/s41588-019-0385-z (2019).
- 11 Hormozdiari, F. *et al.* Leveraging molecular quantitative trait loci to understand the genetic architecture of diseases and complex traits. *Nat Genet* **50**, 1041-1047, doi:10.1038/s41588-018-0148-2 (2018).

REVIEWERS' COMMENTS

Reviewer #4 (Remarks to the Author):

The current version of the manuscript is a major improvement in both clarity and thoroughness.

The new analyses concerning GWAS and molecular trait integration are satisfactory and sound. The integrative analysis results (such as FOCUS) for NUP50 are convincing.

As for my previous concerns:

major concern 1: I consider this fully addressed.

major concern 2: I consider this fully addressed.

In context of the author's reply, I have a recommendation: I would clarify in the main text that they meta analyze via Stouffer method - and if they used METAL in particular.

major concern 3: This is fully addressed and the table 1 makes a more convincing case for the NUP50 analysis.

minor concern 4: I'm confused by the authors' reference to "Figure 2b".

"Figure 2" in the current article is a single Manhattan plot (see last paragraph of authors' response to my major concern 1)

Under the assumption that this was meant to be "Fig 1b" and its rework; and given the expanded explanation: I consider my concern addressed.

minor concern 5: I consider this fully addressed.

minor concern 6: I consider this fully addressed.

minor concern 7: I consider this fully addressed.

minor concern 8: I consider this fully addressed.

I have two additional recommendations for the authors to consider:

recommendation 1: the "supplemental methods" text still includes a reference to Giambartolomei's coloc, but is not used in the current version of the manuscript. Maybe

recommendation 2: Table 1: naming a column "Causal" sounds overreaching. I suggest moderating this to a more cautious statement and use a column name like "Causal evidence" instead.

Sincerely,

Alvaro Barbeira

Reviewer #4 (Remarks to the Author):

The current version of the manuscript is a major improvement in both clarity and thoroughness.

The new analyses concerning GWAS and molecular trait integration are satisfactory and sound. The integrative analysis results (such as FOCUS) for NUP50 are convincing.

We thank the reviewer for this positive appreciation of our revision.

As for my previous concerns:

major concern 1: I consider this fully addressed.

major concern 2: I consider this fully addressed.

In context of the author's reply, I have a recommendation: I would clarify in the main text that they meta analyze via Stouffer method - and if they used METAL in particular.

We agree with the reviewer and we then clarify in the text what method was used to perform meta-analysis on the TWAS.

major concern 3: This is fully addressed and the table 1 makes a more convincing case for the NUP50 analysis.

minor concern 4: I'm confused by the authors' reference to "Figure 2b".

"Figure 2" in the current article is a single Manhattan plot (see last paragraph of authors' response to my major concern 1)

Under the assumption that this was meant to be "Fig 1b" and its rework; and given the expanded explanation: I consider my concern addressed.

As the reviewer assumed, this was a typographical error in the point by point response. We apologize and should have referred to Figure 2, rather than Figure 2b.

minor concern 5: I consider this fully addressed.

minor concern 6: I consider this fully addressed.

minor concern 7: I consider this fully addressed.

minor concern 8: I consider this fully addressed.

I have two additional recommendations for the authors to consider:

recommendation 1: the "supplemental methods" text still includes a reference to Giambartolomei's coloc, but is not used in the current version of the manuscript. Maybe

We agree with the reviewer and since FOCUS was used instead of coloc for fine mapping, the reference to coloc was removed.

recommendation 2: Table 1: naming a column "Causal" sounds overreaching. I suggest moderating this to a more cautious statement and use a column name like "Causal evidence" instead.

We agree with the reviewer that "Causal" is an overstatement. We replaced it with "Causal evidence" in table 1.